# Subcellular proteomics reveals a blueprint for endosymbiont integration in trypanosomatid *Angomonas deanei*

Michael Hammond [1,2], Ľubomíra Chmelová[3,9], Natascha A. van Geelen-Kuenzel [4], Anay K. Maurya [4,10], Eden R. Ferreira[5], Vanesa Puente[1], Lawrence Rudy Cadena[4], Kristína Záhonová [1,3,6,7], Adam Dowle [8], Jeremy C. Mottram [5], Eva C. M. Nowack [4] ✉, Julius Lukeš [1,2] ✉ & Vyacheslav Yurchenko [3] ✉

The acquisition of endosymbionts is a fundamental process that has driven the evolution of eukaryotes. The tree of life is filled with cases of internalised prokaryotes that have become integrated into their hosts, often forming mutually beneficial relationships. The trypanosomatid *Angomonas deanei* is one such case, harbouring a single β-proteobacterial endosymbiont. This symbiotic relationship is highly advanced, as evidenced by the identification of host-encoded proteins that are targeted to the bacterium and control its division. To deeper understand this integration, we performed an in-depth subcellular proteomic analysis to determine the compartmental localisation of both host and endosymbiont proteins. Our analysis resolved over 5,000 host proteins and over 400 endosymbiont proteins. We used this rich dataset to identify several novel host-encoded proteins targeted to the bacterium, and validated our predictions using genetic manipulations and microscopy. By mapping the localised enzymatic repertoire, we were able to shed light on metabolic interplay between the two organisms. We confirmed an energetic basis for the previously observed association between the host's glycosomes and its endosymbiont, and discovered an interaction between the endosymbiont and the host's acidocalcisomes. This subcellular proteomic dataset provides a comprehensive foundation for future research into the remarkable process of bacterial integration.

Mitochondria and plastids, the central energy-providing organelles of eukaryotic cells, originated from bacterial endosymbionts that were acquired over one billion years ago and gradually became extensively integrated into the host cell[1,2]. This process involved (i) a massive reduction of endosymbiont genes either through gene loss or replacement of certain endosymbiont proteins by those of the host, often via Endosymbiotic Gene Transfer (EGT), (ii) the emergence of protein translocons allowing for their import into the endosymbiont, (iii) an

[1]Institute of Parasitology, Biology Centre, Czech Academy of Sciences, České Budějovice, Czechia. [2]Faculty of Sciences, University of South Bohemia, České Budějovice, Czechia. [3]Life Science Research Centre, Faculty of Science, University of Ostrava, Ostrava, Czechia. [4]Institute of Microbial Cell Biology, Heinrich Heine University, Düsseldorf, Germany. [5]York Biomedical Research Institute and Department of Biology, University of York, York, UK. [6]Department of Parasitology, Faculty of Science, Charles University, BIOCEV, Vestec, Czechia. [7]Division of Infectious Diseases, Department of Medicine, University of Alberta, Edmonton, AB, Canada. [8]Bioscience Technology Facility, Department of Biology, University of York, York, UK. [9]Present address: Masaryk University, Faculty of Science, Department of Experimental Biology, Brno, Czechia. [10]Present address: LPHI, UMR5294, CNRS, University of Montpellier, Inserm, Montpellier, France. ✉e-mail: e.nowack@hhu.de; jula@paru.cas.cz; vyacheslav.yurchenko@osu.cz

intricate interlinking of host and endosymbiont metabolism, (iv) the establishment of nuclear control over organelle division and segregation, (v) the creation of contact sites between endosymbiont and host organelle membranes, and (vi) the evolution of anterograde and retrograde signalling systems. This complex integration generated a synergistic and homoeostatic system, in which the former host and endosymbiont can no longer be regarded as separate organisms, but rather parts of a novel entity with new cellular and biochemical properties.

In nature, countless more-recently acquired bacterial endosymbionts provide diverse physiological benefits to their hosts, often with notable ecological and economic impact, such as the spheroid bodies from cyanobacteria, which allow diatom *Epithemia* to fix nitrogen, or the sulphate-reducing symbionts of obligatory anaerobe *Anaeramoeba* spp., which interact with host hydrogenosomes[3–6]. Traditionally thought to interact mostly by metabolite exchange, recent studies have revealed certain symbioses progressing this integration much further. Indeed, along with metabolic functions, the endosymbiont-derived chromatophores of *Paulinella chromatophora* and nitroplasts of *Braarudosphaera bigelowii* synchronise their cell division with that of their hosts akin to true organelles[7].

Trypanosomatid flagellates (Euglenozoa: Kinetoplastea) are obligatory parasites of vertebrates, invertebrates, and plants[8,9]. Their notable dixenous representatives (two hosts in their life cycle) are of medical (*Leishmania* and *Trypanosoma*) or agricultural (*Phytomonas*) importance[10,11]. All these lineages, however, have evolved from common ancestors with monoxenous parasites (one host in their life cycle) of insects[12,13].

Trypanosomatid endosymbionts were first described in mosquito-infecting *Trypanosoma culicis*[14] that was later renamed *Strigomonas culicis*[15]. Its bacterium, *Candidatus* Kinetoplastibacterium spp. (Betaproteobacteria: Burkholderiales: Alcaligenaceae) belongs to the same bacterial group as those found in trypanosomatids of the genera *Kentomonas* and *Angomonas*, which together form the subfamily Strigomonadinae[16]. As judged from phylogenetic inferences, endosymbiont acquisition by a common ancestor of Strigomonadinae was a single evolutionary event 40–120 million years ago, marking the start of a long co-evolution process[17,18]. The subfamily Strigomonadinae constitutes one of only two recognised endosymbiotic acquisitions within Trypanosomatidae, the other being *Novymonas esmeraldas* and its multiple copies of endosymbiont *Ca.* Pandoraea novymonadis[19].

The relationships between symbionts and their Strigomonadinae hosts appear to be well-integrated and mutualistic, as evident from their highly reduced genome sizes relative to free-living bacteria[20,21], coordinated cell cycles[22,23], association with host glycosomes[24,25], and established metabolic cooperation alleviating the hosts' dependence on the environmental availability of essential nutrients, such as heme, nucleotides, and certain amino acids[26–28].

Originally isolated from a reduviid bug *Zelus leucogrammus*[29], *A. deanei* infects a wide range of mosquito and blowfly species in nature[30–32]. At least one strain of *Angomonas deanei* can be experimentally deprived of its endosymbiont[33] enabling resolution of factors essential to support this relationship. However, another *A. deanei* strain (ATCC PRA-265, used in this study) has proven incapable of experimental deprivation through all attempted conditions thus far[34]. The establishment of a "toolkit" for genetic manipulation of the *A. deanei* nuclear genome has facilitated investigations of certain molecular underpinnings mediating this endosymbiont relationship[35–37]. An initial proteomic characterisation of isolated endosymbionts has identified seven Endosymbiont-Targeted host Proteins (ETPs) regarded as key candidates in gaining nuclear control over the endosymbiont[24]. Two of these ETPs, namely the dynamin-like protein ETP9 and the mostly intrinsically disordered protein ETP2, were shown to play essential roles in the division of the bacterial endosymbiont[34,38].

Similarly, ornithine cyclodeaminase (originally encoded by an endosymbiont gene that was later transferred to the nucleus via EGT) is targeted to *A. deanei* glycosomes, presumably facilitating proline production in these organelles[24]. It is plausible to suggest that other ETPs and EGTs are yet to be discovered and will be key in exploring the extent and defining features of this endosymbiotic integration.

To gain a more comprehensive perspective on the state of host-endosymbiont interactions in terms of protein targeting, metabolism, and cell biology, we employed subcellular proteomics, resolving over 5000 proteins and assigning 2938 of them to specific cell compartments, subcellular structures, as well as the endosymbiont, further identifying seven new putative ETPs through predictive clustering in *A. deanei*. The enzymatic localisation implies the endosymbiont's metabolic dependence on energy substrates provided by the glycosomes, reflecting their close association. We additionally use this dataset to identify a novel association between the endosymbiont and acidocalcisomes, likely mediating calcium signalling between these compartments.

This localised repository of both *A. deanei* and endosymbiont proteins enhances our knowledge of the relationship between this trypanosomatid and its internal bacterium. Our work also shows the informative power of the subcellular proteomics for hypothesis exploration across new model organisms, particularly those, facilitating endosymbiotic relationships.

## Results
### LOPIT-DC: marker assignment for 21 well-defined cell compartments
*Angomonas deanei* was lysed via nitrogen cavitation and underwent Localisation of Organelle Proteins by Isotope Tagging by Differential Ultracentrifugation (LOPIT-DC), with fractionated distribution of proteins verified by label-free detection gels and western blot analysis (Supplementary Fig. 1). In total, 5796 proteins passed quality thresholds and were present across all four biological replicates. They included 5323 proteins encoded by the host and 473 by the endosymbiont, constituting 51% and 65% of their predicted proteomes, respectively. Analysis of t-SNE (t-distributed Stochastic Neighbour Embedding) representation for proteins, highlighting genome of origin (host or bacterium) demonstrates a single distinct cluster of endosymbiont-encoded proteins populated with several previously identified ETPs[24], with just six endosymbiont-encoded contaminant proteins outside of this cluster (individual inspection of fractional profiles of these proteins shows fractional inconsistency across replicates) (Fig. 1A, left and middle). Sub-localisation of the endosymbiont-encoded proteins revealed a mostly homogenous distribution of its four predicted bacterial compartments (i.e., cytoplasm, periplasm, inner and outer membranes), suggesting the bacterium remained mostly intact upon lysis of *A. deanei* cells (Fig. 1A, right) and informed our designation of a single set of marker proteins to define the endosymbiont for supervised classification (10–20 proteins were used for all marker groups within this dataset) (Supplementary Data 1A).

To assess host cell compartment distribution beyond the selection of antibody markers employed (Supplementary Fig. 1C), we analysed the host-encoded dataset via various annotation pipelines. Proteins with predicted mitochondrial targeting peptides (mTPs) were enriched across one expansive region of the t-SNE plot. Conversely, proteins with predicted signal peptides were enriched within three primary clusters, inspection of which showed canonical proteins of the endoplasmic reticulum (ER), Golgi apparatus, and acidocalcisomes, with each of these clusters also enriched in transmembrane domain proteins (TMDs). Other inspected clusters enriched for TMDs included proteins corresponding to the glycosomes and mitochondrial membranes (Fig. 1B, C).

In total, we curated a list of 351 marker proteins (Supplementary Data 1A), additionally informed by marker lists used for subcellar

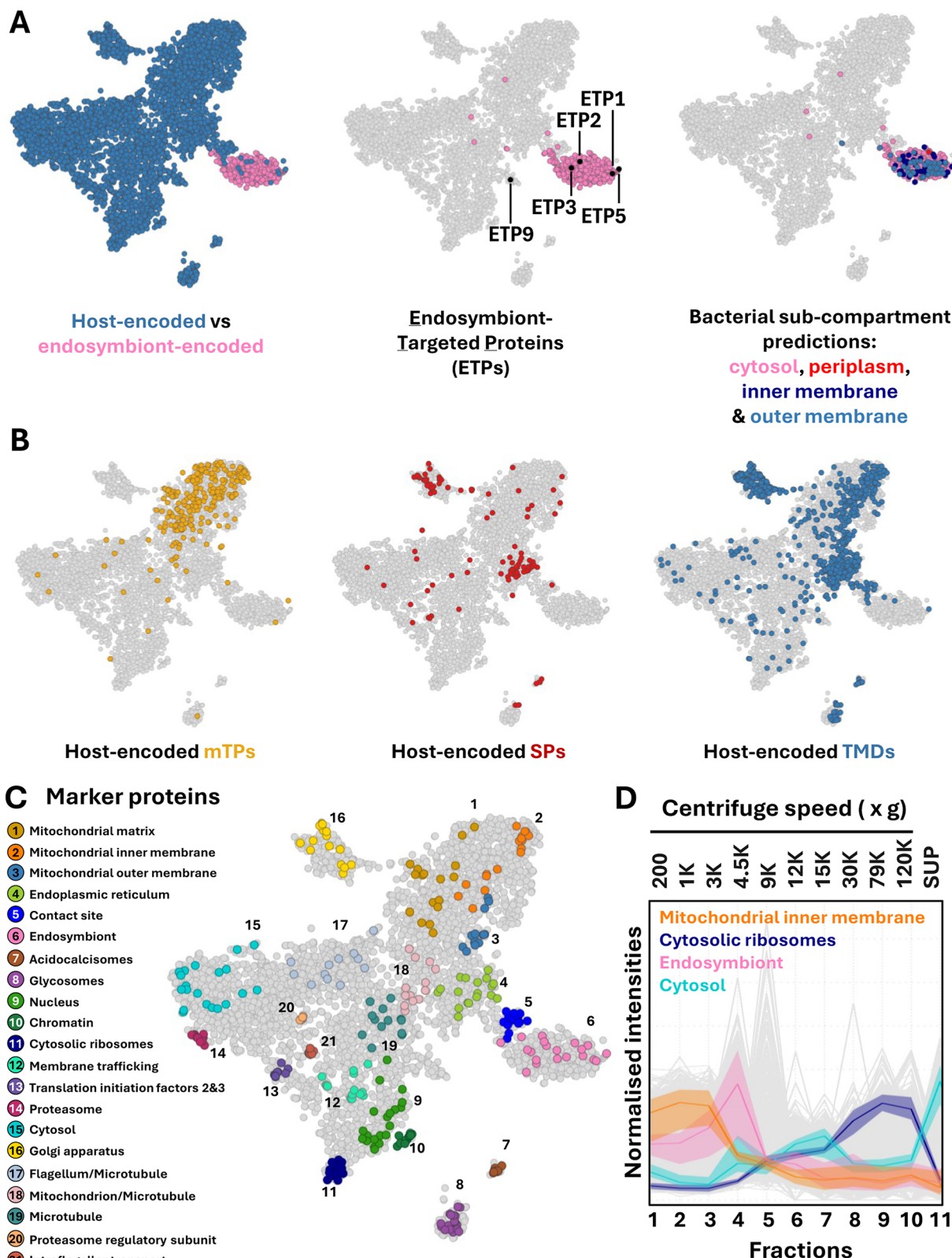

**Fig. 1 | Cell compartments of *A. deanei* illuminated via marker proteins across subcellular proteome. A** t-SNE distribution of host and endosymbiont-encoded proteins, highlighting endosymbiont-targeted proteins (ETPs), along with predicted bacterial sub-compartmental localisation of all endosymbiont-encoded proteins (473). **B** Predicted features across host-encoded proteins (5,323), displaying predicted mitochondrial targeting peptides (mTPs), signal peptides (SPs), and transmembrane domain proteins (TMDs). **C** All 21 marker protein groups shown on t-SNE representation. **D** Fractional distribution of four example marker protein groups across one experimental replicate, showing distinct separation profiles. Central lines represent mean fractional profiles, with shaded band representing ± 1 standard deviation. Relative protein intensities indicated on y axis, with x axis showing centrifuge speeds on top of graph (SUP, supernatant), corresponding to different fractions below, marker protein accessions available in Supplementary Data 1A.

proteomics of *Trypanosoma brucei* and *T. congolense*[39]. This list corresponded to 21 cellular regions (Fig. 1C) that displayed distinct fractional abundance profiles (four examples are shown in Fig. 1D, all profiles are documented in Supplementary Fig. 2). Certain components within specific cell compartments or complexes displayed fractional distinction enabling their sub-designation, for example, allowing us to assign separate markers for the mitochondrial matrix, as well as the inner and outer mitochondrial membranes (Fig. 1C, Supplementary Data 1A). Similarly, the nucleus separated into a chromatin cluster, other soluble components, as well as membranous proteins, while subunits of the proteasome complex showed fractional distinction from the proteasomal regulatory subunits (Fig. 1C). In contrast, proteins of other organelles/structures, including the large and small subunits of cytosolic ribosomes, remained tightly associated, as such we employed single marker sets for each of these compartments/structures.

## Validation of subcellular localisations predicted by LOPIT-DC

Supervised classification predicted 2898 (50%) of the 5796 identified proteins to distinct sub-cellular structures based on scores above support vector modelling thresholds (Fig. 2A), with the remaining proteins classified with lower confidence, but ultimately predicted as 'unknown' (Supplementary Data 1B–D). To verify localisation of the predicted clusters, we tagged 20 proteins that either served as reference markers or were newly assigned to these clusters with enhanced green fluorescent protein (eGFP), using an overexpression system described previously[24,35] (Fig. 2A, Supplementary Data 1E).

Fluorescent signal confined to distinct cellular regions was observed for 13 of the tested proteins. By contrast, two other proteins showed ambiguous patterns, while five other recombinant proteins displayed no signal. The above-mentioned 13 proteins served to verify prediction localisations to the cytosol, nucleus, mitochondrion, glycosomes, Golgi apparatus, and endosymbiont (Fig. 2). Cytosolic marker CAD2222212 (Fig. 2B[1]) and candidate CAD2221863 (Fig. 2B[2]) both show a broadly distributed signal lacking specific enrichment to any other recognisable sub-compartment. Conversely, a nuclear candidate CAD2215914 co-localises entirely with the nuclear DNA (Fig. 2B[3]), while a second nuclear candidate (CAD2220566) is confined to a smaller nuclear region, likely corresponding to the nucleolus (Fig. 2B[4]). Three mitochondrial candidates (CAD2222276, CAD2213008, and CAD2219020) display a tubular signal near the periphery of the cell (Fig. 2B[5–7]), which reflects the structure and position of this single reticulated organelle[35]. Moreover, one of these proteins (CAD2219020) additionally shows enrichment around the kDNA (Fig. 2B[7], Supplementary Fig. 3[7]), a signal reminiscent of the kinetoplast-proximal profile typical for the *T. brucei* proteins of the mitochondrial matrix tagged with GFP[40].

To verify organelles lacking DNA, we employed double tagged cell lines, using the *trans*-Golgi network marker protein Arf-like 1 (Arl1) C-terminally tagged with the V5 epitope[24], for which we observed a co-localisation of the Golgi marker CAD2212931, and the candidate Golgi-associated protein CAD2219791 (Fig. 2B[8,9], Supplementary Fig. 3[8,9]). To visualise the glycosomes, we employed the peroxisomal targeting signal 1 (PTS1 -SKL) fused to the C-terminus of the red fluorescent protein mCherry[35,41,42]. This signal overlapped with that of the PTS1-bearing glycosomal marker CAD2212694, as well as with the signals for both glycosomal candidates, CAD2217526 and CAD2213015, the latter also bearing a PTS1 (Fig. 2B[10–12], Supplementary Fig. 3[10–12]). Two microtubule candidates (CAD2214043 and CAD2217941) showed a faint flagella pattern, however, due to the very low fluorescence signals, their localisations remain ambiguous (Supplementary Fig. 3; Supplementary Data 1E).

To verify the newly assigned group of the host-encoded endosymbiont-localised proteins, we N-terminally tagged representative CAD2214939 and observed the corresponding fluorescence signal

exclusively at the endosymbiont (Fig. 2B[13], Supplementary Fig. 3[13]). We term this protein ETP10 as the newest member of this group of host-encoded proteins identified in a similar manner[24]. Overall, the cell lines described above validate our predictive clustering.

## An expanded list of identified ETPs

Of the 430 proteins confidently predicted to be localised in or at the endosymbiont (Supplementary Data 1B, C), 11 are encoded by the nuclear genome of *A. deanei* (Fig. 3A). This includes four previously identified ETPs [ETP1 (CAD2220707), ETP2 (CAD2221027, for corrected gene model please refer to[38]), ETP3 (CAD2213480), and ETP5 (CAD2216821)[24]], ETP10 (CAD2214939) newly identified via tagging in this study, along with six novel putative ETPs (Fig. 3A[A-F]). Of the latter category, only CAD2214941 (A) and CAD221941 (D) possess functional annotation as a 'myosin heavy chain protein' and 'structural maintenance of chromosomes protein' respectively (Supplementary Data 1F). Two candidates, CAD2215126 (C) and CAD2218418 (F) are predicted to possess TMDs, while one other, CAD2222252 (D) contains an mTP (Supplementary Data 1F).

Phylogenetic distribution of previously identified ETPs within this LOPIT-DC dataset (except for Euglenozoa-wide ETP5) is confined to the Kinetoplastea clade, with ETP1 lacking orthologues in any other species, suggesting its recent emergence in *Angomonas* spp. (Fig. 3B, Supplementary Data 1G). Strigomonadinae-restricted ETP2 and ETP10 are further absent from the divergent genome of *Kentomonas sorsogonicus*[43]. Notably, putative ETPs CAD2214941 (A) and CAD2214943 (B) show sequence similarity and are assigned to the same orthogroup in TriTrypDB, yet, surprisingly, do not show similar distribution patterns amongst trypanosomatids (Fig. 3B). An inspection of their genomic position (on chromosome 4) shows their adjacency to one another, as well as to ETP10 (Fig. 3C). Each of these three genes remains interspersed by one shorter gene, which lacked the necessary peptide coverage for placement into this dataset, likely influenced by their notably shorter length relative to their neighbours (Fig. 3C). All three of these proteins (ETP10, A, B) additionally possess multiple coiled-coil regions, while ETP A and B are further predicted to carry extensive regions of α-helical globular domains (Supplementary Data 1F).

Notably, the previously identified ETP9 (CAD2212698) is predicted neither to the endosymbiont cluster, nor to any other clusters within this LOPIT-DC clustering (Figs. 1A, 3A). Previous studies have characterised ETP9 as a dynamin-like protein, interacting transiently with the outer division-site of the endosymbiont during late-stage division, otherwise, uniquely showing a weak cytosolic signal for the remainder of the cell cycle[34]. Lacking confident designation to a marker-based predictive cluster, we employed a 'marker-less' unsupervised classification, which distributed ETP9 to a small group of 40 mostly hypothetical proteins (Fig. 3A, Supplementary Fig. 4, Supplementary Data 1H). Along with ETP9, these proteins are unified via their prominent fractional intensity at fraction 5 (9,000 × g), suggesting an associated density slightly lower than that of the endosymbiont, which peaks in the preceding fraction 4 (4500 × g) (Fig. 3D). This fractional profile suggests an associated density greater than specific protein complexes (such as the cytosolic ribosomes, Fig. 1D) and other low-density organelles (such as the Golgi apparatus, Supplementary Fig. 2), which sediment primarily in the following fractions. Since the only other investigated protein within this cluster is ETP9's paralogue - a dynamin-related protein (CAD2218610, Fig. 3A), we term this the "dynamin cluster".

## Endosymbiont association with host nucleus and endoplasmic reticulum characterised in 'contact site' cluster

The bacterial endosymbiont of *A. deanei* has previously been documented in spatial proximity to the glycosomes[24], nucleus[22], and ER[44]. This prompted us to investigate whether fractional protein evidence of these associations was present in the analysed dataset. Predictive

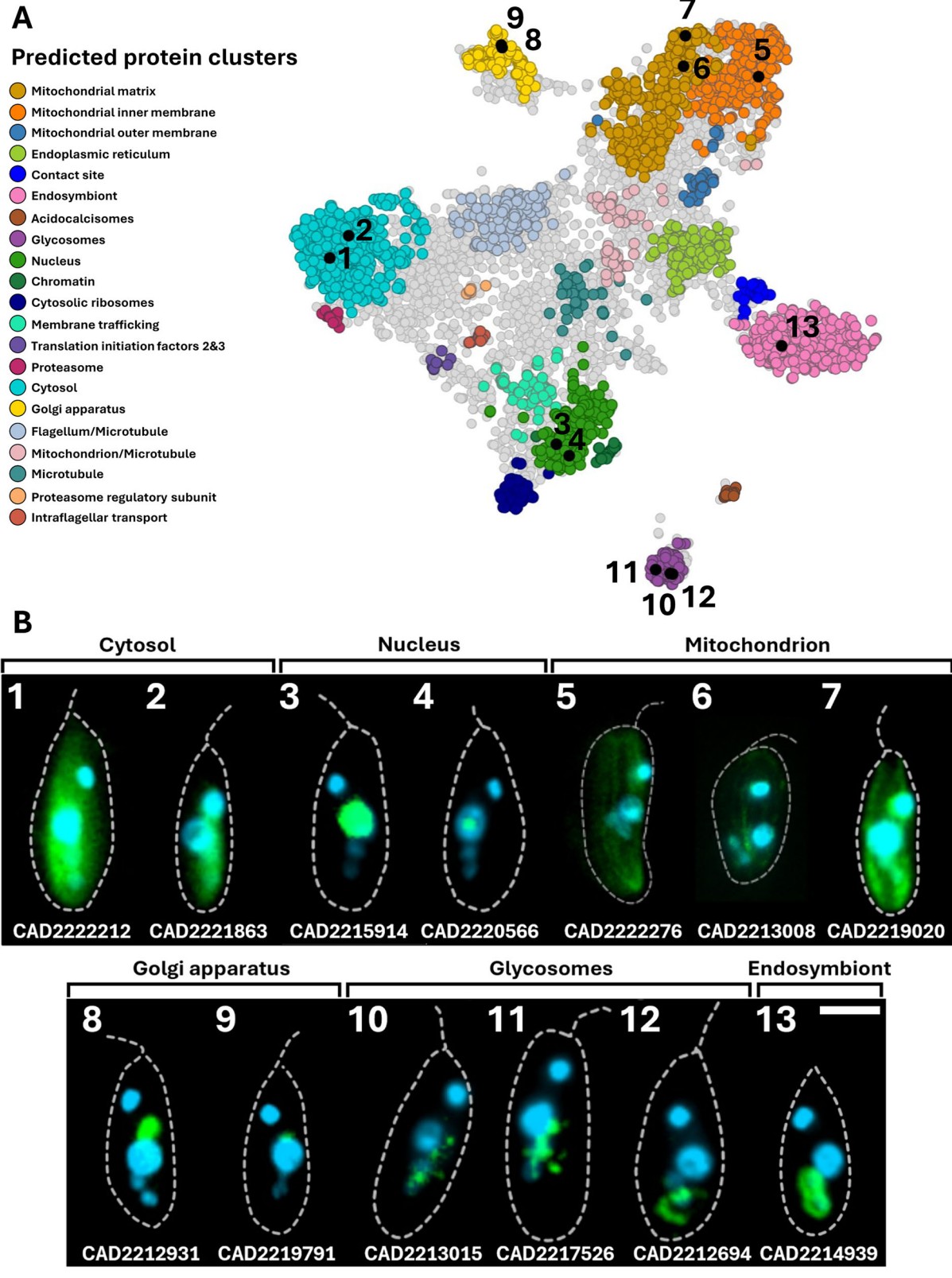

**Fig. 2 | Tagged cell lines of *A. deanei* validate predictive clustering across multiple compartments. A** Protein clusters corresponding to indicated compartments and subcellular structures of *A. deanei*, with host-encoded proteins producing definitive tagged localisations indicated in black. **B** Cell lines generated to express eGFP fused to relevant proteins, analysed by epifluorescence microscopy. Relevant proteins localised to the cytosol (1,2), nucleus (3,4), mitochondrion (5,6,7), Golgi apparatus (8,9), glycosomes (10,11,12), and endosymbiont (13). Protein signal (green) is merged with DNA (blue), with cell outline indicated via dotted line. Scale bar (2.5 μm) applies for all images. Separate channels for appropriate lines, along with additional markers for the Golgi apparatus and glycosomes available in Supplementary Fig. 3. Information on predicted protein localisations available in Supplementary Data 1B, information specifically concerning tagged proteins available in Supplementary Data 1E.

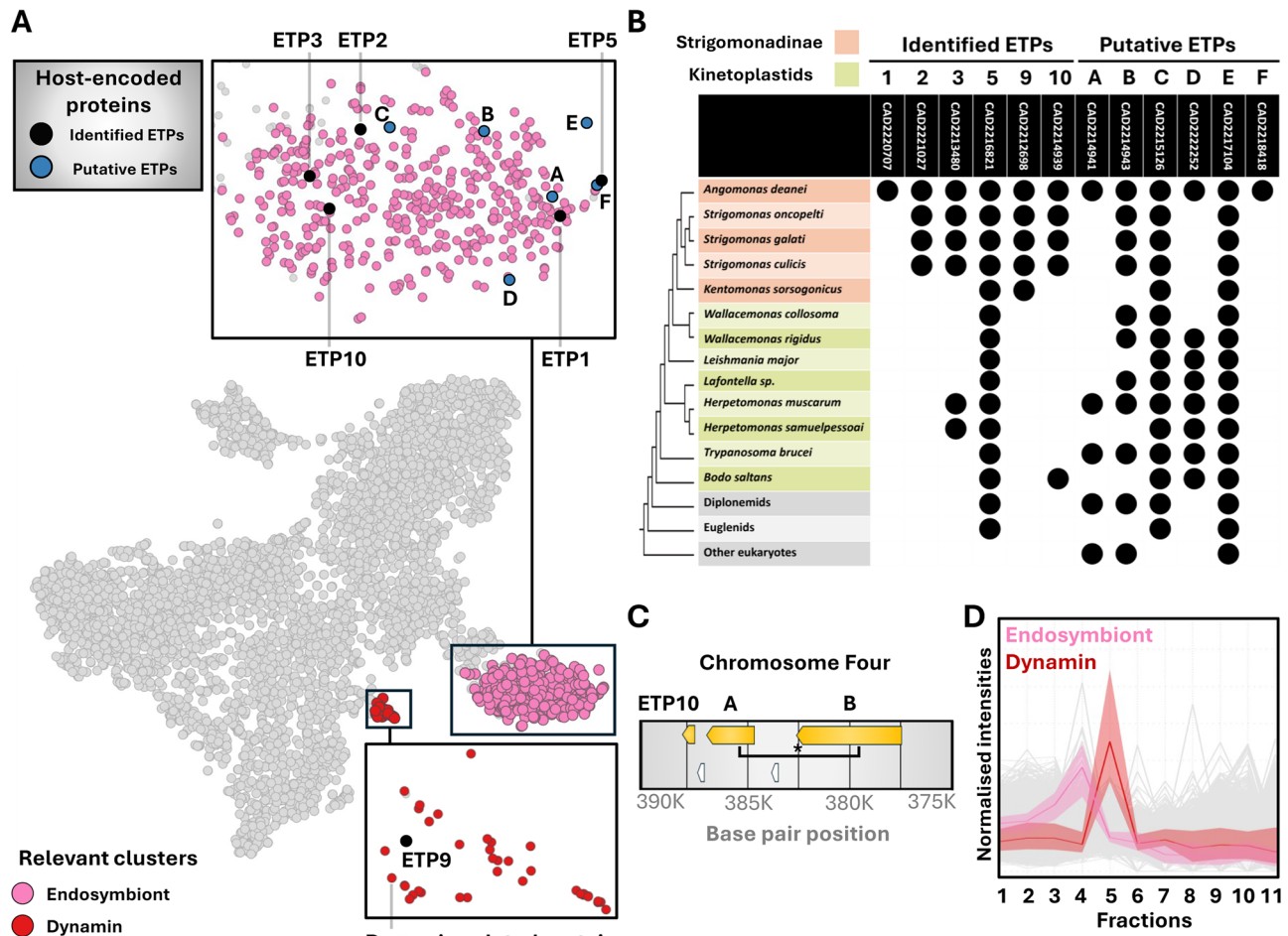

**Fig. 3 | New putative endosymbiont-targeted proteins (ETPs) identified in subcellular dataset. A** Proteins localised via predictive clustering to the endo-symbiont (pink), with host-encoded proteins among these predictions indicated, either as putative or confirmed ETPs. The 'dynamin-cluster' containing ETP9 depicted red. **B** Distribution of ETPs across kinetoplastid and eukaryotic representatives. Relatedness shown via cladogram, with endosymbiont-bearing sub-family Strigomonadinae in orange and other kinetoplastids in lime green. **C** ETPs encoded by adjacent genes shown on genomic locus relative to each other, genes with products not localised in this study are shown in white. Asterisk indicates assignment to same orthogroup (OG6_100282). **D** Fractional distribution of the dynamin cluster relative to the endosymbiont. Central lines represent mean frac-tional profiles, with shaded band representing ± 1 standard deviation. Further information available in Supplementary Data 1F, G.

clustering designates a group of 44 host-encoded proteins, which we term "contact site" (Fig. 4A, blue), exhibiting a fractional profile mir-roring that of the endosymbiont, but with reduced peak intensity (Fig. 4B). This group is enriched for nuclear membrane proteins along with several canonical ER components (Supplementary Data 1I) but shows distinct separation from the 'native' fractional profiles of both the soluble nucleus and ER clusters, of notably lower densities (Fig. 4B), suggesting that these proteins specifically sedimented with the endosymbiont. We suggest that unlike ETPs assigned directly to the endosymbiont cluster, this group of host-encoded proteins are targeted to host organelles, which then directly or indirectly tether to the bacterium and remain attached post cell lysis.

Manual inspection of the contact site cluster by a combination of targeting signal prediction, functional annotation, DeepLOC compart-ment prediction, analysis of orthologues present in *T. brucei* combined with their established localisations allowed assignment of these pro-teins to their "organelles of origin", with approximately half of this group including the nuclear membrane (22), followed by a notable contingent of the ER (17), with a minority (7) appearing to originate from other or unknown cellular regions (Fig. 4C, Supplementary Data 1I). This contact site appears to comprise all nuclear pore complex proteins and is distinct from the clusters of both soluble nuclear

proteins and chromatin components, which are depleted for the TMD-containing proteins (Fig. 1B, Fig. 4B). A separate TMD-enriched cluster of the ER proteins was resolved within this contact site (Fig. 1B, Fig. 4B, C), suggesting that while the entire membranous proteome of the nucleus has seemingly sedimented with the endosymbiont, only a small portion of the ER has remained attached after cell lysis, likely containing proteins immediately adjacent to the endosymbiont. The presence of unidirectional calcium (CAD2220154) and UDP-galactose (CAD2218422) transporters within the ER-assigned component of the contact site specifically suggests the transfer of these solutes from the endosymbiont to the ER (Supplementary Data 1I).

## Acidocalcisomes interact with the endosymbiont in a similar manner to the glycosomes

Glycosomes exhibit a distinct fractional profile, with prominent intensity in fraction 4, representing endosymbiont-associated orga-nelles, while their presence in fractions 5 through 8 reflects the den-sities of the un-associated or 'free' glycosomes sedimenting closer to their presumed native density (Fig. 4D). Accordingly, we demonstrate by fluorescent microscopy a subset of glycosomes in close proximity to the endosymbiont ($Gl_1$), as well as those separated and distributed across the cell ($Gl_2$) (Fig. 4E).

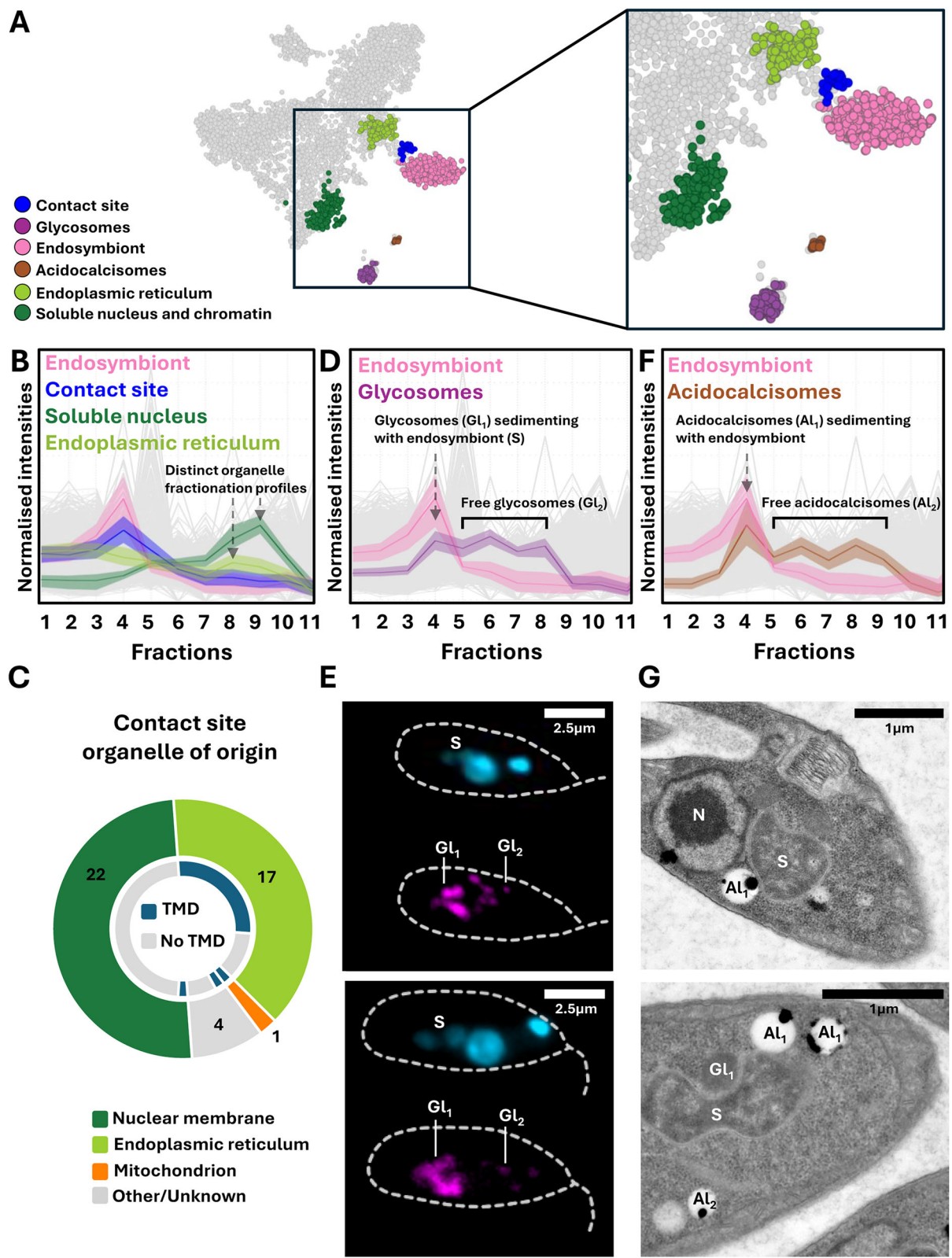

**Fig. 4 | Nucleus, endoplasmic reticulum, glycosomes, and acidocalcisomes all show organelle interaction with endosymbiont. A** t-SNE representation of the endosymbiont, contact site, and ER clusters, with selected fractional profiles (**B**). Central lines represent mean, with shaded band representing ± 1 standard deviation. **C** Manual designations for organelle of origin concerning contact site proteins (Supplementary Data 1I). **D** Glycosomal fractional profile documenting both association and separation with the endosymbiont confirmed by fluorescent microscopy (**E**) of intact cells showing endosymbiont (S), associated (Gl₁) and free (Gl₂) glycosomes. ~500 cells were assessed for representative images. Scale bar (2.5 μm) applies to all fluorescent microscopy images. **F** Acidocalcisomes show similar fractional profile to that of glycosomes, implying selective association with the endosymbiont, confirmed by transmission electron microscopy (**G**), showing acidocalcisomes interacting (Al₁) or free (Al₂) from the endosymbiont. ~50 cells were assessed for representative images. N, nucleus. Scale bar (1 μm) applies to all electron microscopy images.

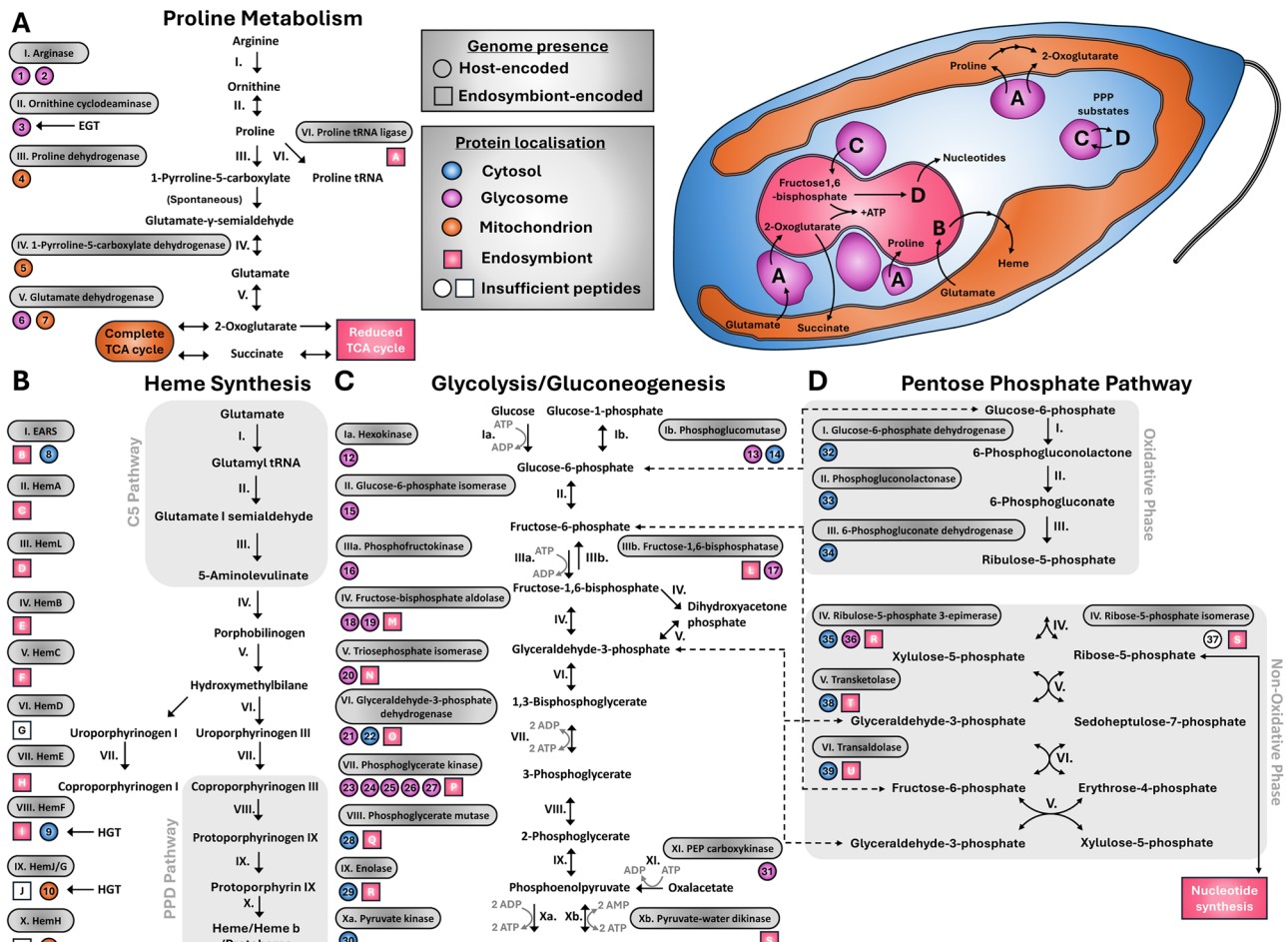

**Fig. 5 | Enzyme distribution shows metabolic interdependency between host and the endosymbiont.** Reconstructions of enzymatic pathways, for proline metabolism (**A**), heme synthesis (**B**) showing C5 and protoporphyrin-dependent (PPD) pathways, glycolysis/gluconeogenesis (**C**), and pentose phosphate pathway (**D**) with metabolite shunts between pathways indicated as dotted lines. ATP conversions for certain reactions and pathway subcomponents are shown in grey. Summary of metabolite transfer between compartments is in top right. Encircled numbers show host-encoded proteins, with endosymbiont-encoded genes indicated by lettered squares, coloured according to proteomic localisation. Endosymbiotic gene transfers (EGT) and horizontal gene transfers (HGT) to the nuclear genome indicated. TCA, Tricarboxylic acid cycle. Accessions for all enzymes available in Supplementary Data 1J.

Another multicopy organelle of interest are the acidocalcisomes, lysosome-related organelles that also show a similar proteomic profile of endosymbiont-associated and endosymbiont-free patterns (Fig. 4F), suggesting an association with the endosymbiont in a similar manner to the glycosomes. This fractional similarity to the glycosomes can be observed via t-SNE spatial resolutions, which depict both the acidocalcisome and glycosome clusters proximal to each other (Fig. 4A).

We confirm this fractional association with transmission electron microscopy images showing a subset of the acidocalcisomes in proximity to the endosymbiont (AI$_1$), as well as those distant from it (AI$_2$) (Fig. 4G). Acidocalcisomes typically serve as acidified reservoirs of calcium, polyphosphates, amino acids, and various heavy metals[45,46]. In the analysed dataset, this cluster is accordingly enriched for transporters of lysine, polyamines, phosphates, calcium, magnesium, potassium, zinc, and other metals (Supplementary Data 1B), suggesting the potential transfer of these substrates from acidocalcisomes to the endosymbiont.

**Enzyme localisation expands on putative metabolite transfer between host and endosymbiont**

Metabolic interplay between the host and endosymbiont represents a key feature of endosymbiotic integration. In the case of *A. deanei*, its bacterium is known to provide various metabolites and cofactors, such as heme, purines, and essential amino acids to the host[27,33,47], while the glycosomes are presumed to supply proline to the endosymbiont[24]. Using the dataset of the current study, we localised core metabolic enzymes encoded by the host (Fig. 5, circles) and endosymbiont (Fig. 5, squares) to highlight putative inter-compartmental exchange of metabolites.

We proteomically validated the glycosomal localisation of EGT-derived ornithine cyclodeaminase[24], which converts ornithine to proline, a key amino acid for energy generation of various insect-stage trypanosomatids[48,49] and endosymbiont localisation of a bacterial proline tRNA ligase (Fig. 5A). Furthermore, we document two glycosomal enzymes necessary to initiate this metabolic synthesis from the precursor amino acid, arginine, and localise succeeding enzymatic steps, allowing us to suggest not only transfer of proline from the glycosome to the endosymbiont, but also the mitochondrion, where it can be subsequently processed to 2-oxoglutarate for tricarboxylic acid (TCA) cycle incorporation, as commonly employed in the endosymbiont-lacking trypanosomatids[50] (Fig. 5A). The presence of a glycosome-localised copy of glutamate dehydrogenase also suggests that 2-oxoglutarate passes to the endosymbiont, since the bacterium is unable to generate this metabolite endogenously (Fig. 5A). While *A. deanei* possesses a complete set of enzymes for the TCA cycle, the endosymbiont's metabolism is reduced almost entirely to the minimal

set of enzymes necessary to power oxidative phosphorylation through NADH, exclusively generated by the conversion of 2-oxoglutarate to succinyl-CoA (succinyl coenzyme A), with additional capacity to produce succinate, and only possessing respiratory complexes I and V (Supplementary Data 1C, J). Lacking the capacity to process this metabolite further, we presume bacterial succinate is then shunted to the mitochondrion for utilisation (Fig. 5A).

*Angomonas deanei* is known to depend on its endosymbiont for heme synthesis[51–53], and accordingly, the pathway of heme synthesis from glutamate shows endosymbiont-exclusive enzymes from step II to VII (Fig. 5B). Step VIII is performed by enzymes from both the bacterium and the host (localised in the cytosol), allowing metabolite transfer to occur either in the form of protoporphyrinogen IX to the cytosol (after step VII), or protoporphyrin IX directly to the mitochondrion (after step VIII), to which the two final host-encoded enzymes of this pathway are confined (Fig. 5B). Notably, the final three host-encoded steps of heme synthesis all represent ancestral bacterial acquisitions via horizontal gene transfer[47,52].

In a typical trypanosomatid, the first seven steps of glycolysis/gluconeogenesis are confined to the glycosomes, with the remaining three enzymes localised in the cytosol[54,55] (Fig. 5C). The endosymbiont genome lacks the first three glycolytic enzymes[56] rendering a functional dependency on the adjacent glycosomes to initiate glycolysis from glucose and perform the energy consuming steps I and IIIa, before fructose 1,6-bisphosphate is putatively transferred to the bacterium (Fig. 5C).

Similar to glycolysis, the first three enzymes (the oxidative phase) of the Pentose Phosphate Pathway (PPP) have been lost from the endosymbiont genome, leaving this bacterium unable to conventionally power the subsequent steps (the non-oxidative phase), which nonetheless are proteomically detected (Fig. 5D). As PPP of the host is primarily cytosolic, with just a single glycosomally-localised enzyme powering step IV in tandem with its cytosolic counterpart (Fig. 5D), the endosymbiont's PPP likely remains dependent on bacterial glycolysis/gluconeogenesis to supply shared metabolites, namely glyceraldehyde 3-phosphate and fructose-6-phosphate (Fig. 5C). In turn, this illuminates the metabolic rationale for retaining the gluconeogenic enzyme fructose-1,6-bisphosphatase (step IIIb, generating fructose-6-phosphate) in the genome, despite conventional gluconeogenesis being non-functional beyond this step in the endosymbiont (Fig. 5C).

The non-oxidative phase of PPP produces ribose-5-phosphate for subsequent nucleotide synthesis (Fig. 5D). The host trypanosomatid is known to depend on its endosymbiont for nucleotide provision[53] and, perhaps unsurprisingly, the host-encoded ribose-5-phosphate isomerase (step IV) has minimal peptide recovery (Fig. 5D). This suggests a near-complete 'ceding' of this key pathway subcomponent from the host to the endosymbiont, though without the noticeable gene loss that is observed in various pathways of the endosymbiont (Fig. 5A, C, D).

## Discussion

Subcellular proteomic studies have provided valuable insights into biology of organelles for various protists, demonstrating, among others, production of one-carbon units and formate in mitochondrion-related organelles[57] or revealing lipid droplets spatially positioned adjacent to endosymbiotic green algae[58]. Here, we employed LOPIT-DC to expand the proteomic perspective on the singular endosymbiont-derived compartment present in *A. deanei*, beyond initial characterisations. Seven ETPs were first identified in the *A. deanei* endosymbiont, screened by mass spectrometry for enriched host-encoded components[24], five of which were also resolved in the current dataset. New insights from the LOPIT-DC analysis include seven new host-encoded proteins that display similar fractional distribution to proteins in the endosymbiont cluster, including four

aforementioned ETPs (Fig. 1A). These proteins represent promising candidates for in-depth investigations to clarify the control exerted by *A. deanei* over its singular endosymbiont, as has been shown for previous ETPs[34,38], further validated by the fluorescent signal observed for the new representative, ETP10 (Fig. 2B[13]). The broad signal of ETP10 observed across the entire endosymbiont is highly reminiscent of ETP1, which sub-localises specifically to the bacterial envelope[24,35]. ETP10, like most other ETPs, lacks functional annotation, but here we show its genomic adjacency to a putative ETPA, functionally annotated as 'myosin motor protein' (Fig. 3C). Consequently, we posit that this position on the chromosome 4, with a putative ETPB being in the neighbouring position, represents a tandem gene array that has undergone radical genome rearrangement in order to account for endosymbiont presence.

Two previously identified ETPs not localised in this study, namely ETP7 and ETP8[24], lack the necessary peptide coverage to be confidently resolved under the current analysis scheme that relies on peptide presence in every fraction across all quadruplicates. It, thus, remains likely that relaxing the proteomic thresholds for inclusion may reveal a greater selection of less-abundant ETP candidates, albeit with reduced localisation confidence.

The LOPIT-DC localisation of ETP9, which was extraneous to the endosymbiont (Fig. 3A), is not entirely unexpected given its temporary association with the bacterium during late-stage endosymbiont division across the cell cycle[34]. Uniquely, ETP9 exhibits no fractional association with other marker proteins employed in this study. Marker-based predictions are inherently limited by existing information available for a given organism and/or its close relatives, and we view ETP9's distinct, reproducible fractionation pattern, shared with a small cohort of other proteins, as indicative of a novel undescribed cellular component for *A. deanei*, rather than an artefact of cell lysis. Within this cluster, only ETP9's paralogue of dynamin-related protein has been investigated to any degree and localised to an apical region of the cell adjacent to the mitochondrion and flagellar pocket[38]. Fractional comparison to other marker-based profiles shows greatest similarity to the microtubule and flagellum/microtubule clusters, which also peak in fraction 5, albeit with notably reduced intensity relative to the dynamin cluster (Supplementary Fig. 4C). As such, we interpret this cluster as a specialised component of microtubule-associated proteins, supported by a selection of motor and microtubule-binding domain proteins amongst the limited available annotations for this cohort (Supplementary Data 1H). An inspection of genes encoding this cluster revealed a strikingly high number (16) of adjacent or near-adjacent (separated by one gene) genes across *A. deanei* chromosomes 2, 4, 5, 9, and 21 (Supplementary Fig. 4D). Of these adjacent genes, eight are further assigned to matching orthogroups via TriTrypDB, which together suggests a series of gene duplications at these loci. We further note that one gene of this dynamin cluster, located on chromosome 4, immediately neighbours putative ETPB, predicted here to the endosymbiont (Fig. 3C, Supplementary Fig. 4D^VI). While validating experimentally this collection of several dozen proteins was beyond the scope of the present study, we speculate that, similar to ETP9, proteins of this cluster are of functional relevance to the endosymbiont and, thus, constitute a promising avenue of investigation in regard to the cytoskeletal modifications known to be produced by the endosymbiont-bearing Strigomonadinae[59].

We note that all ETPs discovered in this organism so far have been of eukaryotic origin (Supplementary Data 1G). The capacity for EGT in *A. deanei* has been documented only for ornithine cyclodeaminase, which is targeted to the adjacent glycosomes instead of the endosymbiont itself, representing an intriguing variation on the previously posited hypothesis on the necessity of protein targeting back to the endosymbiont before functional gene transfer can occur[60]. While *A. deanei* appears to have developed the necessary mechanisms for targeting proteins to the endosymbiont, mass gene transfer from the

bacterium to the host nucleus is yet to be documented. However, further investigation of protein phylogeny and experimental tagging is needed before we can conclusively rule out the presence of any ETPs of bacterial origin in this organism.

The metabolic factors leading to glycosomal association of *Ca.* Kinetoplastibacterium crithidii with its host remain incompletely defined. Transfer of proline to the proline-auxotrophic bacterium has been predicted[24] and is supported here with the localisation of proline tRNA ligase within the endosymbiont (Fig. 5A). Other metabolic inferences have been made from genome analysis of this endosymbiont, noting the loss of genes encoding several enzymatic steps across core metabolic pathways for the TCA cycle, glycolysis, and the PPP[26,27,61]. Here, we complement these predictions with evidence that such modified pathways are proteomically present within the bacterium (Fig. 5), as opposed to being functionally redundant and lacking expression. Our enzymatic localisations ultimately predict an endosymbiont that is dependent on energy-rich substrates, including 2-oxoglutarate and fructose-1,6-bisphosphate, which can both be conveniently generated and supplied by the glycosomes (Fig. 5A, C).

We compare these findings with the endosymbiont *Ca.* Pandorea novymonadis of the related trypanosomatid *Novymonas esmeraldas*, regarded as a more recent acquisition[53], which, importantly, does not appear to be associated with the glycosomes of this host[19]. This bacterium is equally auxotrophic for proline, having lost this pathway since its divergence from the free-living relatives, but not via EGT to its host that can presumably produce proline in the cytosol[61]. A partial gene loss is also noted across core metabolic pathways of this endosymbiont, including the oxidative phase of the PPP. The losses across glycolysis of hexokinase (step Ia), phosphofructokinase (IIIa), phosphoglycerate mutase (VIII), and a pyruvate kinase (X) suggest an even more baroque transfer of metabolites between the bacterium and presumably the glycosomes, with a similar net ATP generation for the endosymbiont as that of *A. deanei*[62].

In contrast to *A. deanei*, the symbiont of *N. esmeraldas* retains a complete TCA cycle with complexes I, II, IV, and V for oxidative phosphorylation[62]. The respiratory chain of the *A. deanei* endosymbiont is restricted to just complexes I and V[63], rendering the bacterium critically dependent on NADH likely generated from processing exogenous 2-oxoglutarate to power its limited oxidative phosphorylation (Fig. 6). As such, we propose that glycosomal energy provision for 2-oxoglutarate likely served as a critical contributor in the eventual endosymbiont association amongst members of the Strigomonadinae. Previous metabolic studies have demonstrated the endosymbiont stimulating *A. deanei* respiration by unknown means, while also showing a critical dependency of the trypanosomatid on complex II for initiating this process (with complex I being dispensable)[63]. In this context, the bacterial processing of 2-oxoglutarate to succinate and return of this metabolite to the host, as postulated here (Fig. 5), represents a plausible mechanism for such respiratory stimulation. Key questions to answer for this metabolic arrangement include the specific transporters used for 2-oxoglutarate and succinate translocation across multiple membranes, as well as the specific electron shuttle employed by *Ca.* Kinetoplastibacterium spp., which has lost the ability to synthesise ubiquinone, yet nonetheless respires independently of *A. deanei*[63].

Established associations between the endosymbiont and the nucleus, ER, as well as the glycosomes can all be proteomically reconstructed via fractional comparisons generated in this work (Supplementary Fig. 2). We note that previous endosymbiont extractions[24], in which cells were sonicated followed by gradient centrifugation, did not produce definitive protein evidence for these associations, though we noted among 'contaminant' host-encoded proteins of that study CAD2219332, which we predict here to the glycosomes, and CAD2219478, classified (albeit with lower confidence) to the contact site (Supplementary Data 1K). We interpret the harsher

purification conditions of the previously reported extraction procedure, in contrast to the nitrogen cavitation employed here, as less conducive to preserving these sensitive interactions with the endosymbiont.

We additionally use the current dataset to demonstrate a new organelle association between the endosymbiont and a subset of acidocalcisomes, which can also be documented by electron microscopy (Fig. 4). Moreover, an analysis of previous literature shows numerous examples of *A. deanei* acidocalcisomes imaged adjacent to the endosymbiont[22,23,59,64], which likely escaped notice due to the dispersed nature of this multicopy organelle across the cell.

While glycosomal proximity to the bacterium has traditionally been presumed to represent a key feature of metabolic integration with the endosymbiont (Fig. 5)[24], the functional interplay between the acidocalcisomes and this bacterium are less intuitive. In *T. cruzi*, the acidocalcisomes supply calcium to the mitochondrion to stimulate and regulate energy production[65] and, additionally, merge with the contractile vacuole to mediate osmotic regulation[66]. While localised to the ER in mammals, the calcium channel inositol 1,4,5-triphosphate receptor is localised to the acidocalcisomes of *T. cruzi*, *T. brucei*[67], and *A. deanei* (Supplementary Data 1B). In the abovementioned members of the genus *Trypanosoma*, the acidocalcisomes coordinate calcium release for mitochondrial signalling to mediate cell growth, differentiation and infectivity[67]. The signalling systems between *A. deanei* and its endosymbiont remain mostly undefined but, based on the current dataset, we can postulate presence of a putative signalling channel supplying calcium to the endosymbiont by the acidocalcisomes. In turn, calcium is returned to the host through the ER via a unidirectional calcium importer localised to the contact site (Fig. 6). While non-specific porins are predicted to the outer endosymbiont membrane, specific calcium transporters in its inner membrane are yet to be identified in the bacterial genome (Supplementary Data 1C), though we also consider the possibility of specific ETPs being employed to mediate such a role.

A subcellular analysis of *A. deanei* ultimately demonstrates that its intimate relationship with the endosymbiont involves the participation of dozens of host-encoded proteins through a complex combination of organelle contact sites and specific ETPs involved in metabolic and signalling coordination. Our work demonstrates the informative power of LOPIT-DC to reveal complex molecular interactions across an individual cell. The dissected trypanosomatid represents a highly suitable experimental model to study the molecular underpinnings of well-integrated endosymbiosis. The generated dataset shall also serve as a repository to explore, which factors related to control, signalling and metabolism have precipitated such an indispensable association.

## Methods

### Cultivation and validation

*Angomonas deanei* Carvalho ATCC PRA-265 (first described as *Crithidia deanei*) isolated from *Zelus leucogrammus* (Hemiptera, Reduviidae)[29] was cultivated at 28 °C in Schneider's *Drosophila* medium supplemented with 10 μg/ml hemin (both from Sigma-Aldrich/Merck, Darmstadt, Germany), 50 units/ml penicillin, 50 mg/ml streptomycin (both from Biowest, Nuaillé, France) as described elsewhere[68]. Note that presence of abovementioned antibiotics has no effect on the bacterial endosymbiont. Cells were diluted twice a week in fresh medium once they reached a density of $1 \times 10^8$ cells/ml. Species identity was validated as described previously[69,70].

### Sample preparation

Cell lysis was performed by nitrogen cavitation as described previously[39,71,72] using a pre-chilled cell disruption vessel 4639 (Parr Instrument Company, Moline, USA). A total of $4 \times 10^9$ *A. deanei* cells from the logarithmic growth phase were washed in 1 × PBS (Phosphate-Buffered Saline), resuspended in the homogenisation medium (0.25 M

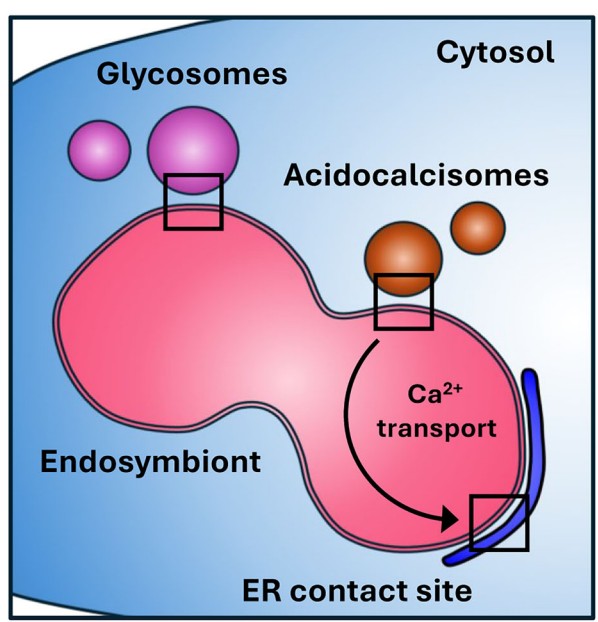

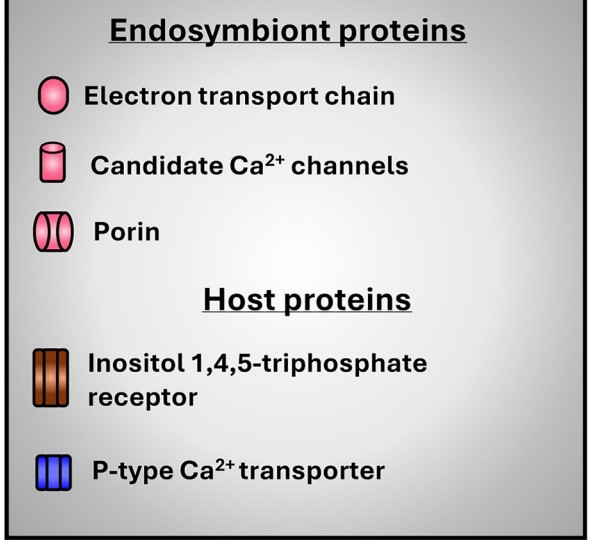

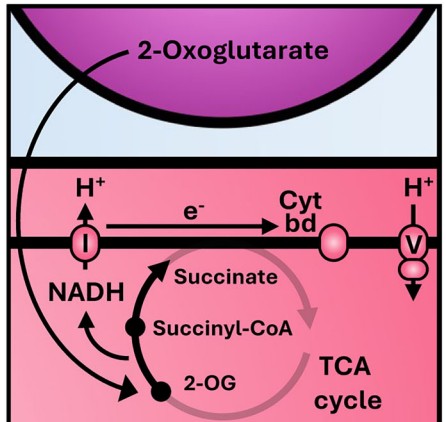

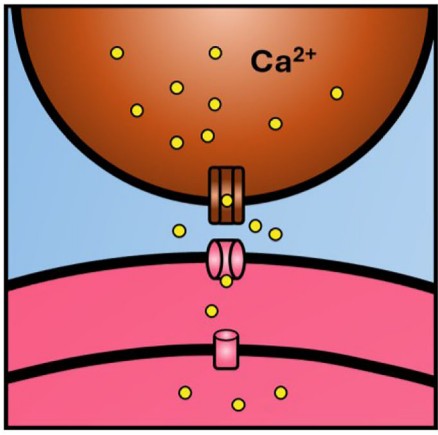

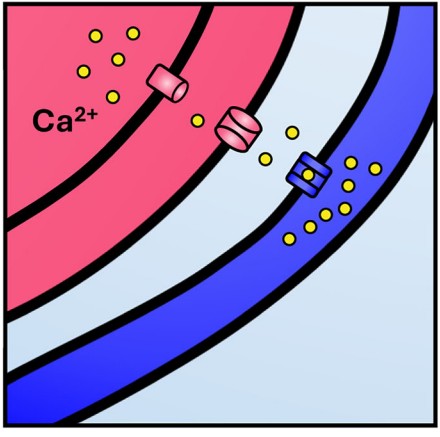

**Fig. 6 | Endosymbiont-interacting compartments of *A. deanei*.** Functional interpretation for glycosomes and acidocalcisomes positioned adjacent to the endosymbiont, showing presumed provision of 2-oxoglutarate (2-OG) to the endosymbiont by glycosomes, as well as calcium transport to and from the endosymbiont through acidocalcisomes and ER. Grey regions of tricarboxylic acid (TCA) cycle represent enzymes lost by the endosymbiont.

sucrose, 10 mM HEPES/KOH pH 7.4, 1 mM EDTA, Halt protease inhibitor cocktail (all from Thermo Fisher Scientific, Waltham, USA)), and disrupted at 1200 psi for 15 min on ice. Lysates were brought to 2 mM of magnesium acetate tetrahydrate and 500 units of Benzonase nuclease (Sigma-Aldrich/-Merck) and kept at room temperature (RT) for 20 min, to remove nucleic acids and reduce sample viscosity, followed by 15 min incubation on ice. Lysates underwent differential centrifugation as described previously[73] (Supplementary Fig. 1A), with fraction 4 modified from 5000 × g to 4500 × g to enable this fraction to be collected in conventional centrifuge Falcon tubes. The experiment was performed in four biological replicates. Replicate protein abundance was assessed via 4-20% Mini-PROTEAN TGX Stain-Free Protein

Gels (Bio-Rad), where 10 µg protein for each fraction was loaded into separate wells, with protein band size referenced against a Precision Plus Protein Strep-tagged recombinant molecular ladder (Bio-Rad) (Supplementary Fig. 1B). Immunoblots were additionally performed on each replicate using the following primary antibodies as indicators for compartment fractional distribution: anti-histone H3 (1:1,000, Abcam, Cambridge, UK, ab18521), anti-EF1α (1:1,000, Merck, 05-235 clone CBP-KK1 [monoclonal]), anti-tubulin β (1:2,000, Sigma-Aldrich/Merck, T0198, clone D66 [monoclonal]), anti-BIP (1:1000, provided by Dr. Bangs[74]), anti-MVK and anti-HMGCS (1:10,000 and 1:5,000 respectively, provided by Dr. González-Pacanowska[75]), anti-OPB (1:20,000)[76] (Supplementary Fig. 1C). Secondary HRP-labelled anti-rabbit and anti-

mouse IgG antibodies were from Promega (Madison, USA); HRP-labelled anti-sheep IgG antibody was from GeneTex (Irvine, USA), all at 1:5,000.

## Digestion, TMT labelling, and high pH reverse phase fractionation

Differential centrifugation fractions were diluted at a 1:3 ratio with 50 mM triethylammonium bicarbonate (TEAB) in 5% sodium dodecyl sulphate and the equivalent of 50 μg of total protein was taken from each fraction for processing. Proteins were reduced with 5.7 mM tris(2-carboxyethyl)phosphine at 55 °C for 15 min, alkylated with 22.7 mM methyl methanethiosulfonate for 10 min at RT, and acidified with 27.5% phosphoric acid precipitated with 7 volumes of 100 mM TEAB 90% (v:v) methanol. Precipitated proteins were captured on S-trap C02-micro columns (ProtiFi, Fairport, USA), washed 5 × with 165 μl 100 mM TEAB 90% (v:v) methanol, and digested with 20 μl 0.1 μg/μl Trypsin/Lys-C mix (Promega) in aqueous 50 mM TEAB at 47 °C for 2 h. Peptides were recovered from the S-traps by centrifugation at 4000 × g for 60 s. Columns were further washed with 40 μl aqueous 0.2% (v:v) formic acid and 40 μl 50% (v:v) acetonitrile:water and eluates were combined. Peptides were dried in a vacuum concentrator and resuspended in 50 μl aqueous 50 mM TEAB for TMT labelling.

Peptides were labelled using TMT11-131C Label Reagent (Thermo Fisher Scientific) following the manufacturer's protocol, with the exception that half of the TMT reagent (0.4 mg) was used per fraction. Samples were combined post labelling, dried in a vacuum concentrator, resuspended in 100 μl water, and loaded onto the 1260 Infinity II LC System (Agilent, Santa Clara, USA) equipped with a Waters XBridge 3.5 μm C18 column (2.1 × 150 mm) (Thermo Fisher Scientific). Separation used gradient elution of solvents A (0.1% ammonium hydroxide) and B (acetonitrile containing 0.1% of ammonium hydroxide). The flow rate was 200 μl/min; the column temperature was 40 °C. The linear multi-step gradient profile for the elution was: 5–35% B over 20 min, 35–80% B over 5 min, the gradient was followed by washing with 80% solvent B for 5 min before returning to initial conditions and re-equilibrating for 7 min prior to subsequent injections. Eluant was collected at 1 min intervals into the protein LoBind tubes (Eppendorf, Hamburg, Germany). Peptide elution was monitored by UV absorbance at 215 and 280 nm. Fractions were pooled across the UV elution profile to give 12 fractions for LC-MS/MS acquisition. Peptide fractions were dried in a vacuum concentrator before reconstituting in 20 μl aqueous 0.1% (v:v) trifluoroacetic acid.

## Liquid chromatography-tandem mass spectrometry (LC-MS/MS)

Fractionated TMT-labelled peptides were loaded onto a Vanquish Neo nano UHPLC equipped with an Acclaim PepMap 100 Å C18 trap (5 μm, 1 × 5 mm) and an Easy-Spray PepMap Neo nano C18 analytical column (2 μm, 75 μm × 500 mm) (all Thermo Fisher Scientific). Separation used gradient elution of solvents A (0.1% formic acid) and B (80% acetonitrile containing 0.1% formic acid). The flow rate for the capillary column was 250 nl/min; the column temperature was 40 °C. The linear multi-step gradient profile was: 5–32% B over 70 min, 32–50% B over 15 min, 50–99% B over 2 min, and then proceeded to wash with 99% solvent B for 3 min. The column was returned to initial conditions and re-equilibrated before subsequent injections.

The nanoLC system was interfaced with an Orbitrap Exploris480 mass spectrometer equipped with an EasyNano ionisation source via FAIMS Pro Duo (all Thermo Fisher Scientific). Positive ESI-MS and MS2 were acquired using Xcalibur v. 4.7 (Thermo Fisher Scientific). Instrument source settings specified ion spray voltage at 1800 V and ion transfer tube temperature at 275 °C. The FAIMS CV was set to −45 V. The MS1 spectra were acquired with 120 K resolution, scan range of m/z 350–1200, AGC target, and other settings left at default. Data-dependent acquisition was performed in a top speed mode using 1 s

cycle, selecting the most intense precursors with charge states 2–5. Dynamic exclusion was performed for 45 s post precursor selection with a tolerance of 10 ppm and a minimum threshold for fragmentation set at 5e³. The MS2 spectra were acquired with 45 K resolution; quadrupole isolation 0.7 m/z; HCD collision energy 35%; AGC target, first mass 110 m/z; max fill time 96 msec.

## Database searching and TMT quantification

Peak picking, database searching, TMT reporter ion extraction and quantification were performed using Proteome Discoverer v. 3.1 (Thermo Fisher Scientific). CHIMERYS v. 2.7 (Thermo Fisher Scientific)[77] was used as the search engine set against the custom-built *A. deanei* and *Ca.* Kinetoplastibacterium crithidii proteomes along with common proteomic contaminants. Search criteria were specified as follows: charge, 1-6; mass error, 10 ppm; missed cleavage, max 2; dynamic modifications, oxidation (M); static modifications, TMT6/10/11-plex (peptide N-terminus and K). Searches were run with a strict false discovery rate of 0.01 and filtered to require a minimum of two unique peptides per accepted protein.

## LOPIT subcellular predictions

Relative, normalised, TMT-derived protein abundances across differential centrifugation fractions among the four biological replicates were used as input values for LOPIT subcellular localisation prediction. Data was filtered to require protein quantification in all four biological replicates, and a minimum of two unique corresponding peptides. Data was imported and processed in R, primarily via pRoloc package as detailed in published Bioconductor workflows[78].

The 351 manually curated protein groups were used as a training set for support vector machine (SVM) model with 'svmOptimisation' and svmClassification' functions using pRoloc. Marker proteins were chosen based on their shared fractional distribution patterns across four biological replicates. 100 rounds of five-fold cross-validation were performed to optimise the SVM parameters based on marker protein abundance profiles. The optimal parameters for the SVM classifier were then applied to all proteins in the dataset with a corresponding SVM score ranging from 0 to 1, with 1 being the score of marker proteins. The SVM classifier was then applied to non-marker proteins, with corresponding weights applied to each marker category (Supplementary Data 1D). Each protein was thus classified to one compartment, and any protein whose classification fell below the global median SVM score was reset to 'unknown' while the other half of the dataset was considered "predicted" to its corresponding compartment due to their higher SVM scores. Quantitative separation values were additionally calculated to determine spatial resolution between each predicted cluster (Supplementary Fig. 5).

Unsupervised clustering was also performed, using the K-means (KM) algorithm implemented in the MLearn function from the MLInterfaces package in Rstudio v. 1.78.0. KM generates k-random centroids and includes surrounding data points iteratively such that all data points are included in one of the k clusters and the size of each centroid is minimised. KM clusters were generated with 21 clusters (Supplementary Fig. 5) corresponding to the number of marker groups used in Supplementary Data 1B, C. Cluster 1 and its 40 constituent proteins was ultimately designated as the 'dynamin cluster' (Supplementary Fig. 4A).

All proteins of this dataset underwent annotation via GhostKOALA[79], with host-encoded proteins undergoing targeting signal predictions via SignalP 6.0[80], TargetP 2.0[81], DeepLoc 2.1[82], DeepTMHMM 1.0[83], and BLASTp searches against parasitic *T. brucei* and free-living *Bodo saltans* from the TriTrypDB release 68[84] with e-value cut-off 1e⁻⁵ (Supplementary Data 1B). In certain cases, orthologue localisation was also determined using TrypTag[85] or *T. brucei* LOPIT data[39] (Supplementary Data 1F, H, I). Endosymbiont-encoded proteins underwent DeepLocPro 1.0 compartment predictions[86] and

BLASTp searches against the predicted proteome of free-living *Burkholderia thailandensis* E264 (Supplementary Data 1C). Putative ETP proteins along with newly identified ETP10 additionally underwent analysis via InterPro v. 98.0[87] (Supplementary Data 1F).

### Construction of plasmids

For the generation of plasmids pAdea423, pAdea425, pAdea429, pAdea436, pAdea444, pAdea445, pAdea446, pAdea447, pAdea448, pAdea450, pAdea451, pAdea461, pAdea462, pAdea463, pAdea464, and pAdea465, genes for CAD2219020, CAD2214939, CAD2218596, CAD2222276, CAD2213008, CAD2221863, CAD2220061, CAD2222212, CAD2219791, CAD2221326, CAD2212931, CAD2215914, CAD2220566, CAD2219447, CAD2217941, and CAD2214043 were amplified from *A. deanei* genomic DNA (gDNA) using the primer combinations 3140/3141, 3134/3135, 3207/3208, 3209/3210, 3211/3212, LC01/LC02, LC03/LC04, LC05/LC06, LC07/LC08, LC15/LC16, LC11/LC12, VP01/VP02, VP03/VP04, VP05/VP06, VP07/VP08, and VP09/VP10, respectively (Supplementary Data 2). For the C-terminal eGFP-tagging, these inserts were used to replace the *lacZ* cassette in the 'tagging vector' pAdea235 containing an *egfp* gene immediately downstream of the *lacZ* cassette (Supplementary Fig. 6) employing Golden Gate cloning[24,88]. Similarly, for the generation of plasmids pAdea424, pAdea426, pAdea432, pAdea433, pAdea449, and pAdea452, genes for CAD2219020, CAD2214939, CAD2212694, CAD2217526, CAD2219791, and CAD2212931 were amplified from gDNA using primer pairs 3151/3152, 3145/3146, 3122/3123, 3126/3127, LC09/LC10, and LC13/LC14, respectively. These inserts were used to replace the *lacZ* cassette in the tagging vector pAdea043 with an *egfp* gene immediately upstream of the *lacZ* cassette by Golden Gate cloning resulting in expression cassettes for fusion constructs with N-terminal eGFP-tags. For the construction of plasmid pAdea428, the gene for CAD2218596 was amplified from gDNA using primer pair 3149/3150 and the backbone (pUMA 1467-δ-ama fr 5'-neo^r-gapdh ir-δ-ama fr 3') from pAdea340 (Supplementary Fig. 6) using primer pair 3147/3148. Both fragments were assembled by Gibson assembly as described earlier[24]. For the construction of plasmid pAdea427, the gene for CAD2217526 was amplified from gDNA using primer pair 3130/3131. pAdea235 was digested with *Bsa*I to remove the *lacZ* cassette, which was replaced with the amplified insert using Gibson assembly, resulting in an expression cassette for a fusion construct with C-terminal eGFP-tag. For the construction of plasmids pAdea439 and pAdea443, the genes for CAD2219607 and CAD2213015 were amplified from gDNA using primer pairs 3128/3129 and 3124/3125, respectively. pAdea043 was digested with *Bsa*I to remove the *lacZ* cassette, which was replaced with the amplified insert using Gibson assembly, resulting in expression cassettes for fusion constructs with N-terminal eGFP-tags. All plasmids were verified by sequencing at Microsynth (Balgach, Switzerland) and Eurofins Genomics (Ebersberg, Germany).

### Generation of transgenic cell lines

DNA cassettes excised from plasmids were stably integrated into the *A. deanei* nuclear genome via homologous recombination as described earlier[24]. In brief, $1 \times 10^7$ cells were resuspended in 17–18 μl of the P3 primary cell solution (Lonza, Basel, Switzerland), mixed with 2-6 μg of the linearised plasmid (in 2–3 μl water) and pulsed using transfection programme FP 158 in a 4D Nucleofector (Lonza). Electroporated cells were recovered in growth medium for 6 h at 28 °C before the respective antibiotic(s) were added. Hygromycin B Gold (InvivoGen, San Diego, USA) and G418 (neomycin) (Sigma-Aldrich/Merck) were used at the final concentration of 500 μg/ml. Clonal cell lines were generated by limiting dilution. Genomic DNA of selected clones was isolated using an adapted DNAzol-based protocol (Thermo Fisher Scientific) and clones were verified by a touch-down PCR using a combination of primers that bind outside of the insertion cassette in the genome and/or Phusion PCR using a combination of primers where one primer binds outside the insertion cassette and the other inside (Supplementary Data 2). To verify the localisation of potential Golgi apparatus and glycosomal proteins, the generated cell lines expressing eGFP-tagged Golgi and glycosomal candidate proteins were additionally transfected with an Arl1-V5 and mCherry-SKL constructs, respectively (Supplementary Fig. 6) to assess co-localisation.

### Epifluorescence microscopy, immunofluorescence assay, and transmission electron microscopy

To detect the autofluorescence of fluorescent fusion proteins in *A. deanei*, epifluorescence microscopy was performed as described previously[24]. In brief, log-phase grown cells were fixed with 3-4% formaldehyde and incubated for 10 min at RT in the dark. Fixed cells were washed twice in 1 × PBS, spotted onto poly-L-lysine-coated glass slides, stained with Hoechst 33342 (30 μg/ml final concentration in PBS), and mounted with antifade reagent SlowFade or Prolong Diamond (both Thermo Fisher Scientific). Imaging was performed with an Axio Imager M.1 (Zeiss, Oberkochen, Germany) using an EC Plan-Neofluar 100×/1.30 Oil Ph3M27 objective (Zeiss). Images were analysed with Zen Blue v. 2.5 (Zeiss) and processed with ImageJ2 software[89]. For mitochondrial staining, cells were centrifuged at 7000 × g for 5 min at RT and the cell pellet was resuspended in 1 × PBS supplemented with 10 mM glucose (Thermo Fisher Scientific). The MitoTracker DeepRed FM Dye (Thermo Fisher Scientific) in DMSO was added to the cells to a final concentration of 1 μM and incubated at 28 °C for 30 min. The cells were washed twice in 1 × PBS and proceeded with formaldehyde fixation as above.

The immunofluorescence assay was performed as described earlier[34]. In brief, cells were fixed with formaldehyde, washed three times in 1 × PBS, and spotted onto poly-L-lysine-coated glass slides. Attached cells were permeabilised with 0.2% TritonX-100, washed with 1 × PBS, and blocked for 45 min at RT in 1% blocking solution (albumin bovine fraction V, pH 7.0 (SERVA, Heidelberg, Germany) in 1 × PBS). Next, cells were incubated with mouse anti-V5 primary antibody (ChromoTek/Proteintech, Planegg, Germany) at 1:100 for 1.5 h at RT, washed thrice in 1% blocking solution, and incubated with anti-mouse IgG secondary antibody conjugated to CruzFluor™ 594 (Santa Cruz Biotechnology, Dallas, USA) at 1:100 for 1 h at RT. Cells were washed, stained with Hoechst 33342, mounted with SlowFade Diamond, and imaged as above.

*Angomonas deanei* wildtype cells were prepared for transmission electron microscopy and imaged as described earlier[34].

### Homology searches

Identified and candidate ETPs served as queries in BLASTp v. 2.11.0 +[90] and tBLASTn searches (e-value cut-off 1e^-5) against a custom-built protein database covering the eukaryotic and prokaryotic diversity and trypanosomatid genomes in NCBI, respectively. Open reading frames of genomic hits were extracted and translated in Geneious Prime v. 2025.1.2[91]. All identified hits were searched by BLASTp (e-value cut-off 1e^-5) against the *A. deanei* genome-derived proteome to identify reciprocal best hits, i.e., orthologues. Additionally, *A. deanei* ETPs were aligned by MAFFT v. 7.458[92] with their orthologues from TriTrypDB release 68, profile Hidden Markov Models were prepared, and HMMER v. 3.3 searches were performed (e-value cut-off 1e^-10) in the same protein database. Again, identified hits were searched by BLASTp (e-value cut-off 1e^-5) against *A. deanei* proteome.

### Reporting summary

Further information on research design is available in the Nature Portfolio Reporting Summary linked to this article.

## Data availability

The raw data generated in this study have been deposited to the ProteomeXchange Consortium *via* the MassIVE partner repository

(MSV000098972, doi:10.25345/C55D8NT39) with the dataset identifier PXD067873. Source data are provided with this paper.

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

## Acknowledgements
The authors thank Ingrid Škodová-Sveráková (Comenius University, Bratislava) for helpful discussions. We acknowledge support of the Czech Science Foundation (25-15298S to V.Y and J.L.), European Union's Operational Programme "Just Transition" (CZ.10.03.01/00/22_003/0000003 LERCO to V.Y.), German Research Foundation (SFB1535, project ID 458090666 to E.C.M.N.), Wellcome Trust (221944/A/20/Z to J.C.M.), and PhD fellowships of the Jürgen Manchot Graduate School (MOI IV to A.K.M. and MOI V to L.R.C.). Computational resources were provided by the e-INFRA CZ project 90254 supported by the Czech Ministry of Education, Youth and Sports.

## Author contributions
Conceptualisation (V.Y., J.L., and E.C.M.N.), methodology (J.C.M.), validation (M.H., L.Ch., and E.R.F.), formal analysis (M.H., K.Z., E.R.F., and A.D.), investigation (M.H., L.Ch., N.A.G.-K., A.K.M., E.R.F., V.P., L.R.C., K.Z., and A.D.), resources (V.Y., J.L., E.C.M.N., and J.C.M.), data curation (M.H., A.D., V.Y., J.L., J.C.M., and E.C.M.N.), writing - original draft (M.H., E.C.M.N., and V.Y.), writing - review & editing (all authors), visualisation (M.H., E.R.F., A.D., and K.Z.), supervision (V.Y., J.L., E.C.M.N., and J.C.M.), project administration (V.Y.), funding acquisition (V.Y., J.L., E.C.M.N., and J.C.M.).

## Competing interests
The authors declare no competing interests.
