## [Transparent Peer Review file · Nature Communications]

Subcellular proteomics reveals a blueprint for endosymbiont integration in trypanosomatid *Angomonas deanei*

Corresponding Author: Professor Vyacheslav Yurchenko

Version 0:

Reviewer comments:

Reviewer #1

(Remarks to the Author)

The research paper by Hammond et al. presents a subcellular spatial proteomics study on the trypanosomatid species *Angomonas deanei* using the LOPIT-DC methodology. The paper is well structured, clearly and concisely written, and very nicely illustrated. The authors managed to perform a very highly resolved LOPIT-DC analysis and obtained informative and interesting results. I find the conclusions well explained and grounded, and the paper is almost ready for publication. I have only a few minor suggestions:

INTRO

(1) When the model species is first mentioned, please consider adding information about its host species and how its ancestral form might have encountered its endosymbiotic bacterium. Is it assumed that the endosymbiont originated from the host gut microbiome?

RESULTS

(2) Six endosymbiont-encoded proteins were detected in host-compartment fractions and interpreted as contaminants. Please clarify why these proteins were classified as contaminants and why the possibility of their export to the host cytoplasm was excluded.

(3) Three clusters are mentioned as containing proteins with SPs: ER, Golgi, and acidocalcisomes. However, the glycosome cluster also includes proteins with SPs. Why is it not listed here? As currently phrased, the text gives the impression that only TMD-containing proteins without SPs were present in glycosomes, which is not the case.

(4) Figure 3B: Please consider adding a schematic phylogenetic tree to the left side of the figure. This would make the distribution easier to interpret and to draw some evolutionary inferences.

(5) Regarding the description of newly and previously identified ETPs: was a search for conserved domains performed? To my knowledge, none of the tools GhostKOALA, SignalP, TargetP, or DeepLoc predict domain structure, though such information would be valuable. Please consider including such an analysis if it has not been done. If it has, please specify the tool used and note explicitly whether conserved domains were absent. Additionally, was SignalP the only tool used to predict TMDs? While it can distinguish SPs from TMs, it does not generate a full TM map. Please specify this in the Methods.

METHODS

(6) When mentioning Benzoylase nuclease treatment of the cell lysate, please explain the purpose of this.

(7) In the section describing cluster annotation with markers and protein compartment assignment, please clarify how biological replicates were taken into account. Only proteins detected in all four replicates were considered reliable; however, I assume there was some variation in distribution across fractions. Please specify how this variability was handled.

Reviewer #2

(Remarks to the Author)

Hammond et al., investigated the localization of proteins in the bacterial endosymbiont-harboring kinetoplastid *Angomonas deanei* using spatial proteomics. They successfully combine a variety of approaches to suggest new insights into the cell biology of this symbiosis which include the identification of new ETPs, and additional support for several host organelle-endosymbiont associations. Aspects of the study are validated with fluorescence and electron microscopy, as well as comparative genomics.

The manuscript is well-written. I do not have any major concerns and think that this study deserves to be published. I agree with their approach and their conclusions. My only suggestion is that, at times, purely correlational inferences are stated in a rather definite way. The hypotheses that this study raise require further experimental validation, but this is out of the scope of the present study. I recommend this manuscript for publication in *Nature Communications*.

Minor comments:

L48-56: Include gene losses that were never replaced. Most genes were probably lost rather than transferred to the host nucleus.

L53-54: What's an example of anterograde signaling system in a symbiotic organelle? Protein-import is already mentioned.

L59: The spheroid bodies allow for nitrogen fixation, not simply nitrogen assimilation.

L158: The selected proteins can clearly not be verified conclusively as the next paragraph described by ambiguous or no signal from some of them.

L242-244: The presence of the nuclear pore complex casts some doubt on the interpretation that this cluster corresponds to a contact site. Is proximity in clustering space among clusters sufficiently suggestive of spatial proximity? It seems to me that this is not always the case. Why?

Reviewer #3

(Remarks to the Author)

COMMENTS: Subcellular proteomics reveals a blueprint for endosymbiont integration in trypanosomatid *Angomonas deanei*

What are the noteworthy results?

The authors performed spatial proteomics on *Angomonas deanei* and were able to identify 5,796 proteins in total, with 5,323 proteins encoded by the host and 473 by the endosymbiont. LOPIT-DC was able to identify 21 cellular regions, which were visualised using t-Distributed Stochastic Neighbor Embedding. This included a single distinct cluster of endosymbiont-encoded proteins. Support vector machine analysis was used to classify 2,898 previously unknown proteins to their localisation in this biological system. Notably of the 430 proteins localised to the endosymbiont compartment, 11 were encoded by the nuclear genome of *A. deanei* and therefore could be endosymbiont-targeted host-proteins. Furthermore, the LOPIT-DC profiles gave evidence for multiple organelles interacting with the endosymbiont, including the nucleus, endoplasmic reticulum, glycosomes, and acidocalcisomes, which was confirmed using microscopy.

Will the work be of significance to the field and related fields? How does it compare to the established literature? If the work is not original, please provide relevant references.

Previous studies have aimed to characterise endosymbiont-targeted host-proteins using expression proteomics methods and fluorescence microscopy; however, this paper represents the first instance of subcellular spatial proteomics to gain insight into this biological system on a global scale. This allows new findings regarding endosymbiont-targeted host-protein discovery and organelle interactions.

Does the work support the conclusions and claims, or is additional evidence needed?

The work supports its spatial conclusions by coupling subcellular proteomics with imaging. The enzymatic mapping gives insight into the metabolic system and suggest that modified pathways could be present within the bacterium, however further evidence could be useful to confirm specific mechanistic details.

Are there any flaws in the data analysis, interpretation and conclusions? - Do these prohibit publication or require revision?

The following points could be addressed by the authors to increase clarity; however, they should not prohibit publication.

- i. The authors utilise t-SNE based dimensionality reduction for spatial mapping, however, it could also be informative to include principal component analysis, as it more accurately preserves the global structure of the dataset for easier interpretation.
- ii. The t-SNE and linear profiles show distinct profiles, however it may be useful to include resolution quantitation analysis, such as QSep or F1 scores, to verify the separation of each compartment cluster.
- iii. Perhaps the mitochondrial (matrix, inner and outer) markers could be curated further to avoid overlap, which could enhance resolution of these regions. If overlap cannot be avoided maybe these regions could be combined into one single mitochondrial cluster instead.
- iv. Figure 4-B, D and F would be clearer with labelled y axes, and the scale ruler in the microscopy figures could include a number on the figure as well as in the legend, to enhance readability.
- v. Why were four biological replicates used? Does this impact the resolution compared to three replicates, and is the number of overall identifications affected?
- vi. What was the rationale for changing the spin speed of fraction 4 from 5,000 to 4,500 $\times g$ compared to the original LOPIT-DC protocol?

Is the methodology sound? Does the work meet the expected standards in your field?

The methodology here is sound as it utilises the well-established LOPIT-DC method, described in reference 68. Further,

transmission electron and fluorescence microscopy enhance this dataset by providing further evidence to build upon the support vector machine predictions. The use of this well-established methodology coupled with microscopy-based validation meets the expected standards within the field.

Is there enough detail provided in the methods for the work to be reproduced?

There is enough detail within the methods section to be able to reproduce this protocol. Furthermore, references 67, 68 and 73 provide further detail of the previously established LOPIT technique workflows. Perhaps the following up to date reference could also be included regarding the data analysis:

Hutchings C, Krueger T, Crook OM et al. An updated Bioconductor workflow for correlation profiling subcellular proteomics. *F1000Research* 2025, 14:714 (<https://doi.org/10.12688/f1000research.165543.1>)

Reviewer #4

(Remarks to the Author)

The dataset presented in the submitted manuscript is technically strong but the work is largely descriptive and, in my view, does not bring new conceptual and mechanistic insights.

Major comments

1. The central advances (expanded ETP set, contact-site clusters, metabolic and Ca²⁺-signalling models) are inferred from co-fractionation profiles, localisation, and pathway reconstruction. Only one newly proposed ETP is experimentally validated, and there is essentially no functional validation of candidate ETPs or contact-site components to demonstrate roles in endosymbiont division, positioning, or host–symbiont homeostasis.
2. The claims of specific nuclear envelope/ER–endosymbiont tethers and an acidocalcisome–endosymbiont Ca²⁺ circuit are not supported by direct evidence of physical contacts. Co-sedimentation and proximity in microscopy are insufficient to firmly establish the proposed structures
3. The manuscript frequently frames the findings as providing “hallmarks” or a “blueprint” for endosymbiont integration, yet all data derive from a single species and life stage, with no matched comparative proteomic analysis. The broader evolutionary extrapolations therefore feel overstated.

Version 1:

Reviewer comments:

Reviewer #1

(Remarks to the Author)

The authors have comprehensively addressed my comments. All my recommendations were considered and adequately incorporated into the revised manuscript. IMHO, the concerns raised by the other reviewers also appear to have been satisfactorily resolved. I therefore recommend the revised manuscript for publication.

Reviewer #3

(Remarks to the Author)

The authors have mostly dealt with my original comments well. The only point where I disagree is inclusion of PCA plots. tSNE plots do not maintain the structure of the data. It is good practice to include PCA plots LOPIT-DC data in the manuscript - perhaps the authors would consider adding them to the supplemental data.

Reviewer #1

The research paper by Hammond et al. presents a subcellular spatial proteomics study on the trypanosomatid species *Angomonas deanei* using the LOPIT-DC methodology. The paper is well structured, clearly and concisely written, and very nicely illustrated. The authors managed to performed a very highly resolved LOPIT-DC analysis and obtained informative and interesting results. I find the conclusions well explained and grounded, and the paper is almost ready for publication.

Thank you very much for the positive evaluation of our work. We appreciate it very much!

I have only a few minor suggestions:

INTRO

(1) When the model species is first mentioned, please consider adding information about its host species and how its ancestral form might have encountered its endosymbiotic bacterium. Is it assumed that the endosymbiont originated from the host gut microbiome?

We have added the following sentence and relevant citations to the Introduction: “Originally isolated from a reduviid bug *Zelus leucogrammus*, *A. deanei* infects a wide range of mosquito and blowfly species in nature.” As for the second part of the question, the Reviewer is right, it is assumed that the endosymbiont originated from the host gut microbiome. Yet, the exact mechanism, by which these osmotrophic parasites successfully acquired internal bacteria remains unknown. The closest known relatives of the *Kinetoplastibacteria* endosymbionts are bacteria of the genus *Taylorella*. These are pathogens populating the urogenital tract of horses; some species have intracellular stages. We would prefer not to add unnecessary speculations to the text, but can do so if the Editor and Reviewer insist.

RESULTS

(2) Six endosymbiont-encoded proteins were detected in host-compartment fractions and interpreted as contaminants. Please clarify why these proteins were classified as contaminants and why the possibility of their export to the host cytoplasm was excluded.

We have added the following clarification to the text that now reads: “(individual inspection of these proteins shows fractional inconsistency across replicates)”. Justification text along these lines was originally present but got lost in edits, so thank you for pointing this out.

To elaborate further, here, we also include fractional profiles for these six proteins below across all replicates with the endosymbiont profile, and any host compartment profile that proteins are visually close to in the t-SNE. The fractional profiles of these endosymbiont-encoded proteins do not

strongly resemble host compartments, and typically in one or two replicates they resemble the endosymbiont profile, leading us manually to classify these proteins as ‘misbehaving’ (furthermore, they do not get predicted by SVM to any compartments because of this fractional dissimilarity).

(3) Three clusters are mentioned as containing proteins with SPs: ER, Golgi, and acidocalcisomes. However, the glycosome cluster also includes proteins with SPs. Why is it not listed here? As currently phrased, the text gives the impression that only TMD-containing proteins without SPs were present in glycosomes, which is not the case.

We modified the text that now reads as follows: “signal peptides were enriched within three primary clusters”, which better reflects our justification here. Within the glycosome “island” of the t-SNE, there were just 5 predicted signal peptides among a total of 137 proteins. Furthermore, these signal peptides were on the periphery of the cluster, indicating slight fractional dissimilarity increasing their likelihood as contaminants, rather than being dispersed more homogeneously throughout a cluster as documented for the ER, Golgi, and acidocalcisomes.

(4) Figure 3B: Please consider adding a schematic phylogenetic tree to the left side of the figure. This would make the distribution easier to interpret and to draw some evolutionary inferences.

Thank you for this suggestion! A schematic phylogenetic tree was added to Fig. 3B.

METHODS

(5) Regarding the description of newly and previously identified ETPs: was a search for conserved domains performed? To my knowledge, none of the tools GhostKOALA, SignalP, TargetP, or DeepLoc predict domain structure, though such information would be valuable. Please consider including such an analysis if it has not been done. If it has, please specify the tool used and note explicitly whether conserved domains were absent.

Thank you for this suggestion. We have employed InterPro analysis for putative ETPs and ETP10. We have also further commented on shared coiled-coil and alpha-helical globular domains between tandem genes of this selection in the Results.

Additionally, was SignalP the only tool used to predict TMDs? While it can distinguish SPs from TMs, it does not generate a full TM map. Please specify this in the Methods.

Additionally, DeepTMHMM was used to predict TMDs. This mention may have gotten lost as well in the edits, thank you for pointing this out. We added this information.

(6) When mentioning Benzonase nuclease treatment of the cell lysate, please explain the purpose of this.

We added the following clarification to the text: “...to remove nucleic acids and reduce sample viscosity”.

(7) In the section describing cluster annotation with markers and protein compartment assignment, please clarify how biological replicates were taken into account. Only proteins detected in all four replicates were considered reliable; however, I assume there was some variation in distribution across fractions. Please specify how this variability was handled.

We added the following clarification to the text: “Marker proteins were chosen based on their shared fractional distribution patterns across four biological replicates.”

Indeed, some variation is expected even in replicates performed with identical conditions. As an example of how variability was handled concerning predictions, the candidate endosymbiont-localised protein that we chose for tagging (CAD2214939.1, ETP10) strongly resembles the fractional profile of the endosymbiont in three of four replicates but is somewhat dissimilar in replicate one.

This dissimilarity contributes to its relatively low prediction confidence (a score of 0.58 out of 1, Suppl. Table 1, Tab E). Ultimately however, this protein is still predicted to the endosymbiont, and even in replicate one, the profile still resembles the endosymbiont profile ‘pattern’, which motivated us to investigate this protein further through tagging.

Reviewer #2

Hammond et al., investigated the localization of proteins in the bacterial endosymbiont-harboring kinetoplastid *Angomonas deanei* using spatial proteomics. They successfully combine a variety of approaches to suggest new insights into the cell biology of this symbiosis which include the identification of new ETPs, and additional support for several host organelle-endosymbiont associations. Aspects of the study are validated with fluorescence and electron microscopy, as well as comparative genomics.

The manuscript is well-written. I do not have any major concerns and think that this study deserves to be published. I agree with their approach and their conclusions. My only suggestion is that, at times, purely correlational inferences are stated in a rather definite way. The hypotheses that this study raise require further experimental validation, but this is out of the scope of the present study. I recommend this manuscript for publication in Nature Communications.

Thank you very much for the positive evaluation of our work. We appreciate it very much. We have toned down several statements to avoid this “rather definitive way”.

Minor comments:

L48-56: Include gene losses that were never replaced. Most genes were probably lost rather than transferred to the host nucleus.

Corrected as suggested. The revised text reads: “...(i) a massive reduction of endosymbiont genes either through gene loss or replacement of certain endosymbiont proteins by those of the host...”

L53-54: What’s an example of anterograde signaling system in a symbiotic organelle? Protein-import is already mentioned.

An example would be light-inducible nucleus-encoded sigma factors that are targeted into the plastids of *Arabidopsis thaliana* as anterograde signals. Under light stimulation, they can externally activate a plastid-encoded bacterial-type RNA polymerase to transcribe plastid photosynthesis genes (doi:10.1038/s41467-022-35080-0). Putting this clarification would not fit into the text stylistically, so we would prefer to leave it as it is, if the Reviewer and Editor agree.

L59: The spheroid bodies allow for nitrogen fixation, not simply nitrogen assimilation.

Corrected as suggested.

L158: The selected proteins can clearly not be verified conclusively as the next paragraph described by ambiguous or no signal from some of them. Contrast with others.

We agree with the Reviewer. The wording of this sentence was ambiguous. For clarity, we deleted it entirely.

L242-244: The presence of the nuclear pore complex casts some doubt on the interpretation that this cluster corresponds to a contact site. Is proximity in clustering space among clusters sufficiently suggestive of spatial proximity? It seems to me that this is not always the case. Why?

The Reviewer is correct that proximity among clusters or co-fractionation profiles by themselves are not sufficient to suggest biological contact amongst the organelles, as this could represent a coincidence of two compartments having the same density, which would then co-fractionate without physical tethering. Equally, tethering after cell lysis could represent an experimental artifact, where cell lysis may cause compartments to attach to one another then co-fractionate, also not reflecting a biological reality (cell lysis is more likely to disrupt existing attachments, but this former possibility cannot be excluded).

In the specific case mentioned by the Reviewer (nuclear pore and contact sites), we were additionally informed by the previously published observations for this organism showing the endosymbiont forming attachments with the nucleus and ER among other organelles. It is unlikely that all the proteins of this cluster are genuinely forming contacts with the endosymbiont; many may represent proteins imbedded in or attached to the nuclear envelope membrane, which interact with the endosymbiont *via* a specific contact site. The text was revised to better reflect this and now reads: “We suggest that unlike ETPs assigned directly to the endosymbiont cluster, this group of host-encoded proteins are targeted to host organelles, which then directly or indirectly tether to the bacterium and remain attached post cell lysis”.

Reviewer #3

The authors performed spatial proteomics on *Angomonas deanei* and were able to identify 5,796 proteins in total, with 5,323 proteins encoded by the host and 473 by the endosymbiont. LOPIT-DC was able to identify 21 cellular regions, which were visualised using t-Distributed Stochastic Neighbor Embedding. This included a single distinct cluster of endosymbiont-encoded proteins. Support vector machine analysis was used to classify 2,898 previously unknown proteins to their localisation in this biological system. Notably of the 430 proteins localised to the endosymbiont compartment, 11 were

encoded by the nuclear genome of *A. deanei* and therefore could be endosymbiont-targeted host-proteins. Furthermore, the LOPIT-DC profiles gave evidence for multiple organelles interacting with the endosymbiont, including the nucleus, endoplasmic reticulum, glycosomes, and acidocalcisomes, which was confirmed using microscopy.

Previous studies have aimed to characterise endosymbiont-targeted host-proteins using expression proteomics methods and fluorescence microscopy; however, this paper represents the first instance of subcellular spatial proteomics to gain insight into this biological system on a global scale. This allows new findings regarding endosymbiont-targeted host-protein discovery and organelle interactions.

The work supports its spatial conclusions by coupling subcellular proteomics with imaging. The enzymatic mapping gives insight into the metabolic system and suggest that modified pathways could be present within the bacterium, however further evidence could be useful to confirm specific mechanistic details.

The following points could be addressed by the authors to increase clarity; however, they should not prohibit publication.

Thank you very much for the positive evaluation of our work. We appreciate it very much!

i. The authors utilise t-SNE based dimensionality reduction for spatial mapping, however, it could also be informative to include principal component analysis, as it more accurately preserves the global structure of the dataset for easier interpretation.

A PCA plot of predictions is shown below for reference. Its resolution gives a stronger impression of mitochondrial co-fractionation similarity shared with the endosymbiont, something that is not visually apparent in the t-SNE plot, and only becomes noticeable through fractional comparison, or inferred from inspection of a minority of proteins found in the ‘contact site’ cluster. It also exemplifies the fractional distinction of the ‘dynamamin’ cluster (22) from the rest of the proteome.

Nevertheless, the separation of certain fractional clusters is visually impaired, such as, for example, the acidocalcisomes/glycosomes (7/8), mitochondrion and ER (1/2/3/4), Golgi and IFTs (13/16/21). For this reason, we would prefer to stick to the t-SNE representation, if this is acceptable by the Reviewer and Editor.

ii. The t-SNE and linear profiles show distinct profiles, however it may be useful to include resolution quantitation analysis, such as QSep or F1 scores, to verify the separation of each compartment cluster.

This is a good suggestion, thank you! The QSep scores for the predicted 21 clusters are now included as a new Suppl. Fig. 5.

iii. Perhaps the mitochondrial (matrix, inner and outer) markers could be curated further to avoid overlap, which could enhance resolution of these regions. If overlap cannot be avoided maybe these regions could be combined into one single mitochondrial cluster instead.

We have made some adjustments to the mitochondrial marker proteins (Fig. 1C), which has served to minimise visual overlap in predictions, particularly between the mitochondrial inner and outer membrane (Fig. 2A).

Avoiding all visual overlap without over-curating the marker set remains difficult since these clusters are fractionally very similar, and cross-interactions between certain proteins of these compartments are expected in specific cases.

iv. Figure 4-B, D and F would be clearer with labelled y axes, and the scale ruler in the microscopy figures could include a number on the figure as well as in the legend, to enhance readability.

Modified as suggested.

v. Why were four biological replicates used? Does this impact the resolution compared to three replicates, and is the number of overall identifications affected?

The simple answer is the more the merrier. Specifically, between 101-278 proteins are 'lost' in identification by employing four replicates rather than three (specifically: 204, 232, 278 and 101 shared proteins are not present in replicates one, two, three and four respectively, see Venn diagram below). However, given the high yield of total proteins (5,795) that are still present from combining all four replicates, we view this as an acceptable loss for better quality predictions.

Importantly, having an extra replicate improves the reliability of predictions for proteins present in all four replicates, particularly in cases where one replicate behaves notably different from the others (see ETP10 above in response to the Reviewer 2's question). Another good example concerns acidocalcisomes and glycosomes, which are distinct clusters with four replicates but visually merge and become harder to distinguish with just three replicates.

It is of course possible for others to assemble and inspect all these other combination plots from our raw data for additional analyses.

vi. What was the rationale for changing the spin speed of fraction 4 from 5,000 to 4,500 ×g compared to the original LOPIT-DC protocol?

It was done for pragmatic reasons to enable this fraction to be collected in conventional centrifuge Falcon tubes. This clarification was now added to the text.

Reviewer #4

The dataset presented in the submitted manuscript is technically strong but the work is largely descriptive and, in my view, does not bring new conceptual and mechanistic insights.

We humbly disagree with a note on new conceptual and mechanistic insights (see below).

Major comments

1. The central advances (expanded ETP set, contact-site clusters, metabolic and Ca²⁺-signalling models) are inferred from co-fractionation profiles, localisation, and pathway reconstruction. Only one newly proposed ETP is experimentally validated, and there is essentially no functional validation of candidate ETPs or contact-site components to demonstrate roles in endosymbiont division, positioning, or host–symbiont homeostasis.

Functional validation and characterisation for proteins of interest is our next priority *via* follow-up studies. The organism under study still remains difficult to work with in a high-throughput manner. For instance, the successful tagging validation for a dozen proteins, already represents an enormous experimental undertaking and expenditure of time. We also feel that the different insights of this paper (new ETPs, contact site proteins, new organelle interactions) deserve their own studies to be properly explored. To cite a handling editor of this manuscript (Dr. E. White), “We will not require experimental validation as requested by reviewer #4”.

2. The claims of specific nuclear envelope/ER–endosymbiont tethers and an acidocalcisome–endosymbiont Ca²⁺ circuit are not supported by direct evidence of physical contacts. Co-sedimentation and proximity in microscopy are insufficient to firmly establish the proposed structures.

We understand that this is not an endpoint of the research into endosymbiont contact sites. Rather, the presented data dramatically narrows down the prospective list of candidate proteins from the entire organelle proteomes of each compartment to a handful that are fractionally distinct from their

organelle of origin. These proteins can now be functionally characterized further for their interactions with bacteria.

3. The manuscript frequently frames the findings as providing “hallmarks” or a “blueprint” for endosymbiont integration, yet all data derive from a single species and life stage, with no matched comparative proteomic analysis. The broader evolutionary extrapolations therefore feel overstated.

Echoing also comments of the Reviewer 2, we have toned down several statements in the manuscript. This particularly concerns the word “hallmark” that can be perceived as an overstatement. Concerning the word “blueprint” (something which acts as a plan, model, or template for others), we believe it is OK in this sense. It is used only twice throughout the text. Although, we are ready to tone it down more if the Editor deems it necessary.

Reviewer #1

The authors have comprehensively addressed my comments. All my recommendations were considered and adequately incorporated into the revised manuscript. IMHO, the concerns raised by the other reviewers also appear to have been satisfactorily resolved. I therefore recommend the revised manuscript for publication.

Thank you very much for positive evaluation of our work.

Reviewer #3

The authors have mostly dealt with my original comments well. The only point where I disagree is inclusion of PCA plots. tSNE plots do not maintain the structure of the data. It is good practice to include PCA plots LOPIT-DC data in the manuscript - perhaps the authors would consider adding them to the supplemental data.

We agree with this point. The PCA plot is now presented as new a panel (A) in Supplementary Figure 5.